# Duplicated ribosomal protein paralogs promote alternative translation and drug resistance

Mustafa Malik Ghulam[1], Mathieu Catala[1], Gaspard Reulet[2], Michelle S. Scott[2] & Sherif Abou Elela [1] ✉

Ribosomes are often seen as monolithic machines produced from uniformly regulated genes. However, in yeast most ribosomal proteins come from duplicated genes. Here, we demonstrate that gene duplication may serve as a stress-adaptation mechanism modulating the global proteome through the differential expression of ribosomal protein paralogs. Our data indicate that the yeast paralog pair of the ribosomal protein L7/uL30 produces two differentially acetylated proteins. Under normal conditions most ribosomes incorporate the hypo-acetylated major form favoring the translation of genes with short open reading frames. Exposure to drugs, on the other hand, increases the production of ribosomes carrying the hyper-acetylated minor paralog that increases translation of long open reading frames. Many of these paralog-dependent genes encode cell wall proteins that could promote tolerance to drugs as their translation increases after exposure to drugs. Together our data suggest a mechanism of translation control that functions through a differential use of near-identical ribosomal protein isoforms.

Ribosomes are ribonucleoprotein complexes required for protein synthesis[1,2]. The basic structure of the ribosome is conserved from bacteria to human with increasing complexity in terms of the number and sizes of rRNAs and proteins in higher eukaryotes[2-5]. Ribosomes are viewed as universal machines built for precision and mass production[6-9]. However, this view is being challenged by observations of variations between ribosomal protein genes in terms of their expression, regulatory pathways, and variable incorporation into ribosomes[10-17]. While the concept of the specialized ribosome is debated, the heterogeneity of ribosome composition and regulatory programs is irrefutable. Most eukaryotes have variable amounts of mRNA produced by the different RPGs and mass-spectrometry analyses continue to detect variations in the ribosome populations extracted from different tissues and growth conditions[10,18].

In the yeast *S. cerevisiae*, ribosomal protein genes (RPGs) are mostly produced from independently regulated duplicated genes (Supplementary Fig. 1)[18]. Out of the 137 yeast RPGs, 118 are duplicated and 94 have introns that must be removed to produce the mRNA[18]. Paralogous ribosomal protein pairs are generally more than 95% identical. However, deleting individual paralogs gives different phenotypes, suggesting functional specialization. The origin of this proposed paralog specialization remains unclear and actively debated[19-21].

Here we systematically assess *S. cerevisiae* paralogs with respect to their expression levels and amino acid differences. For the five most disparate pairs, in terms of expression or sequence, we used gene conversion to create simplified "homogenized" strains, in which each individual duplicated RPG was expressed both from its natural locus and as a replacement of its paralog and tested them for growth under different conditions. Overall, our results indicate that duplicated ribosomal protein genes regulate drug resistance through paralog-specific effects on translation.

---

[1]Département de microbiologie et d'infectiologie, Faculté de médecine et des sciences de la santé, Université de Sherbrooke, Sherbrooke, QC, Canada.
[2]Département de biochimie et génomique fonctionnelle, Faculté de médecine et des sciences de la santé, Université de Sherbrooke, Sherbrooke, QC, Canada. ✉e-mail: sherif.abou.elela@usherbrooke.ca

## Results

### Gene conversion of duplicated ribosomal protein genes reveals paralog subfunctionalization

Comparing the expression level of Yeast's 29 duplicated ribosomal protein gene pairs with significant differences in both regulatory and coding sequence using RNA sequencing and mass spectrometry, we

identified five gene pairs: one 40S and four 60S RPG pairs, that are differentially expressed (Fig. 1a–c and Supplementary Fig. 1a, b). Most of these genes exhibit differences in the regulatory sequence controlling their expression (<70% similar) (Supplementary Fig. 1b), which explain why they produce different amounts of RNA and proteins (Fig. 1b, c). In general, paralogs producing most RNA also produced

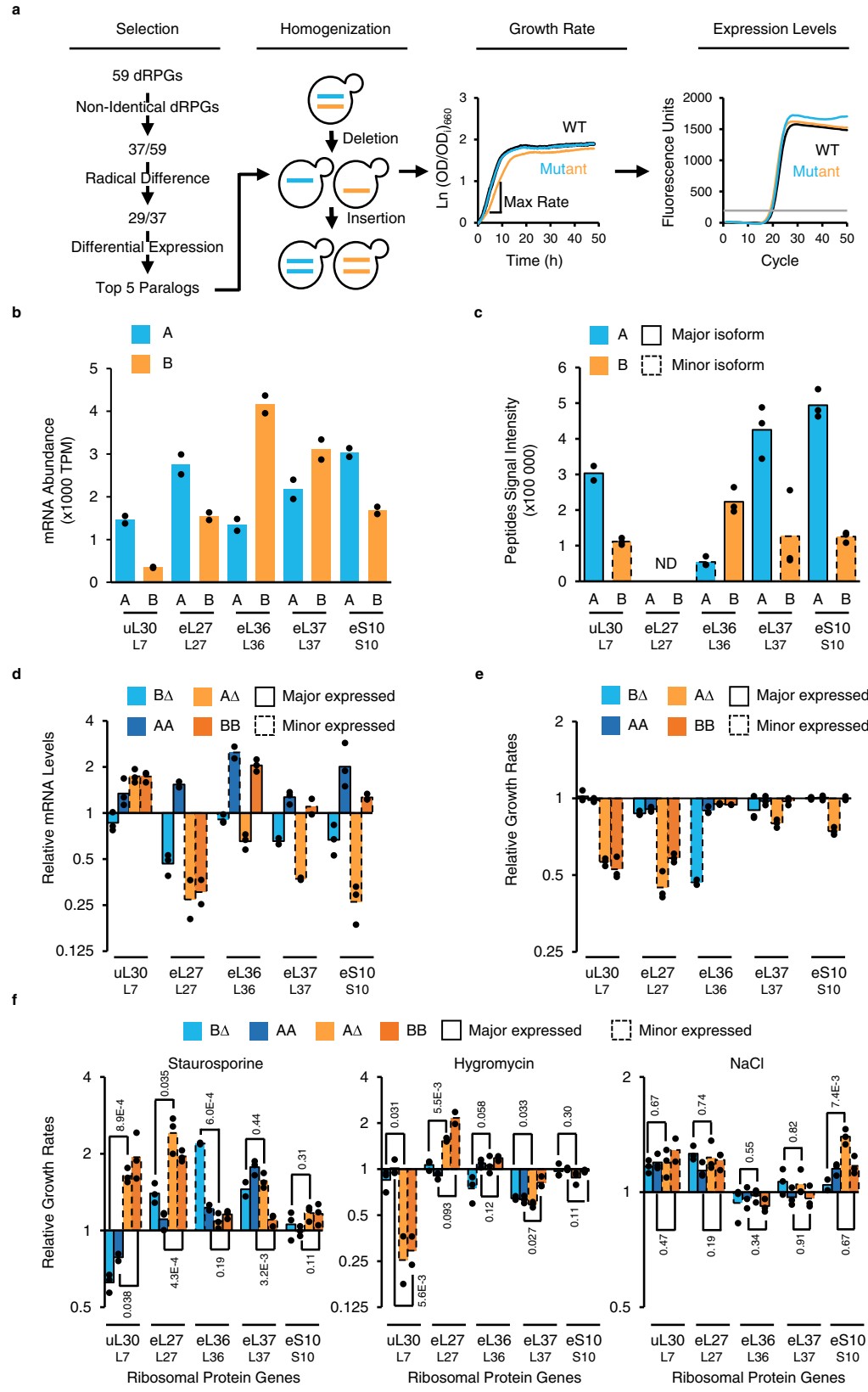

**Fig. 1 | Homogenization of yeast ribosomal proteins identifies dose independent paralogs subfunctionalization. a** Gene conversion strategy for "homogenization" of ribosomal protein genes. Genes showing paralog-specific response to different conditions were selected for homogenization. Homogenized strains were created by replacing one paralog locus including all regulatory sequences, introns and UTRs, with sequence of the other. Cells carrying one or two copies of a single paralog were examined for growth in liquid media and maximum growth rates were calculated for each strain. Duplicated ribosomal protein gene expression levels were examined by RT-qPCR to identify dose-dependency of growth defects. **b** RNA abundance in transcripts per million (TPM) of each paralog obtained by RNA sequencing $n = 2$ biologically independent samples. Universal and conventional names of each protein are shown below. **c** Abundance of protein isoforms generated by each duplicated ribosomal protein gene was determined by Swath Multiple Reaction Monitoring (MRM) identifying the major and minor isoforms. No suitable peptides to reliably distinguish between L27 paralogs were found, hence their peptide signal intensities were not determined (ND). **d** Total RNA generated by each gene pair was detected using primers common to both paralogs in RT-qPCR after deletion or homogenization of duplicated ribosomal protein genes and is shown relative to mRNA detected in wild-type strain. BΔ and AΔ indicate RNA detected in strains lacking the B or A paralog, while AA and BB indicate RNA detected in cells containing two copies of the A or B isoforms respectively. **e** Growth rates of deletion and homogenized strains were determined and shown relative to wild-type. **f** Growth rates of deletion and homogenized strains were determined in media containing staurosporine (3 μg/ml), hygromycin (100 μg/ml) or NaCl (0.9 M) and the effect on growth are shown relative to wild-type treated similarly. Bars shown in (**c**–**f**) represent the means of $n = 3$ biologically independent samples shown as data points, $p$ values from two-tailed unpaired $t$ test are indicated in (**f**).

most protein, except in the case of *eL37/RPL37* paralogs, which are differentially translated[22]. In all cases, one of the two paralogous proteins (the major form) dominated over the other (the minor form) by 3 to 4 times, suggesting that most of the ribosomes are produced from one gene copy (Fig. 1c). Deleting one of the two duplicated ribosomal protein gene (dRPG) copies reduced the abundance of the mRNA generated by the pair, except in the case of *uL30/RPL7* (BΔ and AΔ) and the major copy of *eL36/RPL36* (BΔ, Fig. 1d).

To probe the function of the gene pairs we next used a strategy of gene homogenization whereby we replace one of the paralogs with the other in its natural locus. This holistic approach helped us identify all differences between paralog functions, including those stemming from difference in expression pattern or intervening sequences, and allowed us to distinguish between gene dose and gene type requirement. The results indicated that homogenization of duplicated ribosomal protein genes, except *el27bb/rpl27bb*, does not decrease the overall amount of mRNA produced for each RPG (AA and BB, Fig. 1d). As expected, the deletion of the major paralog, which produces most RPs in the cell, impaired cell growth in rich media (Fig. 1e). This growth defect is restored by the expression of two copies of the minor paralog, except in the case of *uL30* and *eL27/RP27*, where the minor paralog duplication fails to complement the deletion of the major copy (Fig. 1e). This indicates that not all copies of dRPGs could independently support normal growth and that the copy-specific requirement varies between ribosomal proteins.

Since many minor paralogs were dispensable for growth under normal condition, we next examined the effect of gene deletion or homogenization on growth under stress. We exposed our different strains to stresses in the form of exposure to three drugs: Staurosporine, Hygromycin and NaCl, which induce an array of different stresses. As expected, exposure to stress revealed different sets of paralog's copy number-specific effects that were not observed under normal growth conditions and could not be explained by reduced RPG expression or constitutive drug independent decrease in growth rate. For example, deletion of the *eL27* major paralog, which reduced overall eL27 protein expression and inhibited growth under normal conditions, enhanced growth in the presence of staurosporine and hygromycin but not NaCl (Fig. 1f). In contrast, deletion of the minor paralog of *uL30*, which did not inhibit gene expression or cell growth, reduced cell resistance to staurosporine (Fig. 1f). These data indicate that duplicated ribosomal protein genes are not completely redundant and they identify *uL30* as a good model for studying differences between their functions.

### *uL30* paralogs differentially modulate ribosome biogenesis and cell growth

The proteins generated from the duplicated *uL30* ribosomal protein paralogs differ by five amino acids, 4 of which are clustered in the N-terminus, leading to a major difference in the predicted secondary structure of the N-terminal domain (Supplementary Fig. 2b). The

protein featuring the more structured N-terminal domain (uL30A) is more abundant, forming the bulk of the ribosome in the cell, making it more likely to be the housekeeping version of the pair (Supplementary Fig. 2c). Consistently, the expression of this major paralog (*uL30A*) is required for normal growth (Fig. 2a). As expected, the absence of the major paralog (*ul30aΔ* and *ul30bb*) reduced the synthesis of the 60S subunit leading to subunit imbalance, reduction in ribosome abundance and the accumulation of 40S subunit awaiting the 60S subunit, or "half-mers" (Fig. 2c and Supplementary Fig. 2). The subunit imbalance was also confirmed by the decrease in 25S/18S rRNA ratio and the overall decrease in the 25S rRNA and the 60S ribosomal proteins (Supplementary Fig. 2f, g). However, we did not see an overall decrease in the abundance of the mRNA coding for the 60S ribosomal proteins or specific decrease in the uL30 mRNA or protein relative to other 60S proteins (Supplementary Fig. 2e, f). Indeed, despite the defect in the 60S synthesis the total amount of uL30 mRNA and protein produced in *ul30bb* cells was similar or even slightly greater than that detected in ul30aa cells. This indicates that the amount of uL30 proteins is not rate limiting and that the effect observed in *ul30aΔ* and *ul30bb* cells is not caused by decreased production of uL30 proteins in these cells. These data indicate that the major paralog is required for the biogenesis of the 60S subunits, and that the duplication of the chromosomal copy of the minor paralog cannot compensate for the function of the major form even when expressed at similar levels.

### *uL30* minor paralog is required for cell resistance to staurosporine

Surprisingly, deletion of the minor paralog, which does not affect cell growth or ribosome biogenesis, rendered cells sensitive to staurosporine (Figs. 1f, 2b and Supplementary Fig. 3). Cells expressing one or two copies of the A version (*ul30bΔ* and *ul30aa*, respectively) were more sensitive to staurosporine and produced shorter polyribosomes than wild-type cells, but the cells were otherwise normal (Fig. 2b, c, and Supplementary Fig. 2f, g). In contrast, cells expressing the B version (*ul30aΔ* and *ul30bb*) were more resistant than the wild-type to staurosporine but they grew slower under normal growth conditions (Fig. 2a, b and Supplementary Fig. 3). Exposure to staurosporine significantly reduced the size of the polyribosomes in cells expressing uL30A but it had no effect on those expressing uL30B (Fig. 2c, d and Supplementary Fig. 3b). This suggests that uL30 paralogs have opposite effects on cell tolerance to staurosporine resulting in increased sensitivity to the drug when the protein is produced from the major copy and increased resistance when it is expressed from the minor counterpart.

The minor paralog-dependent resistance of staurosporine is not linked to reduced growth rate since cells growing at rates similar to *uL30bb* did not exhibit resistance to staurosporine (Supplementary Fig. 4a). Staurosporine resistance also was not reproduced by the deletion of the translation initiation factors eIF2A or TIF4632, which are

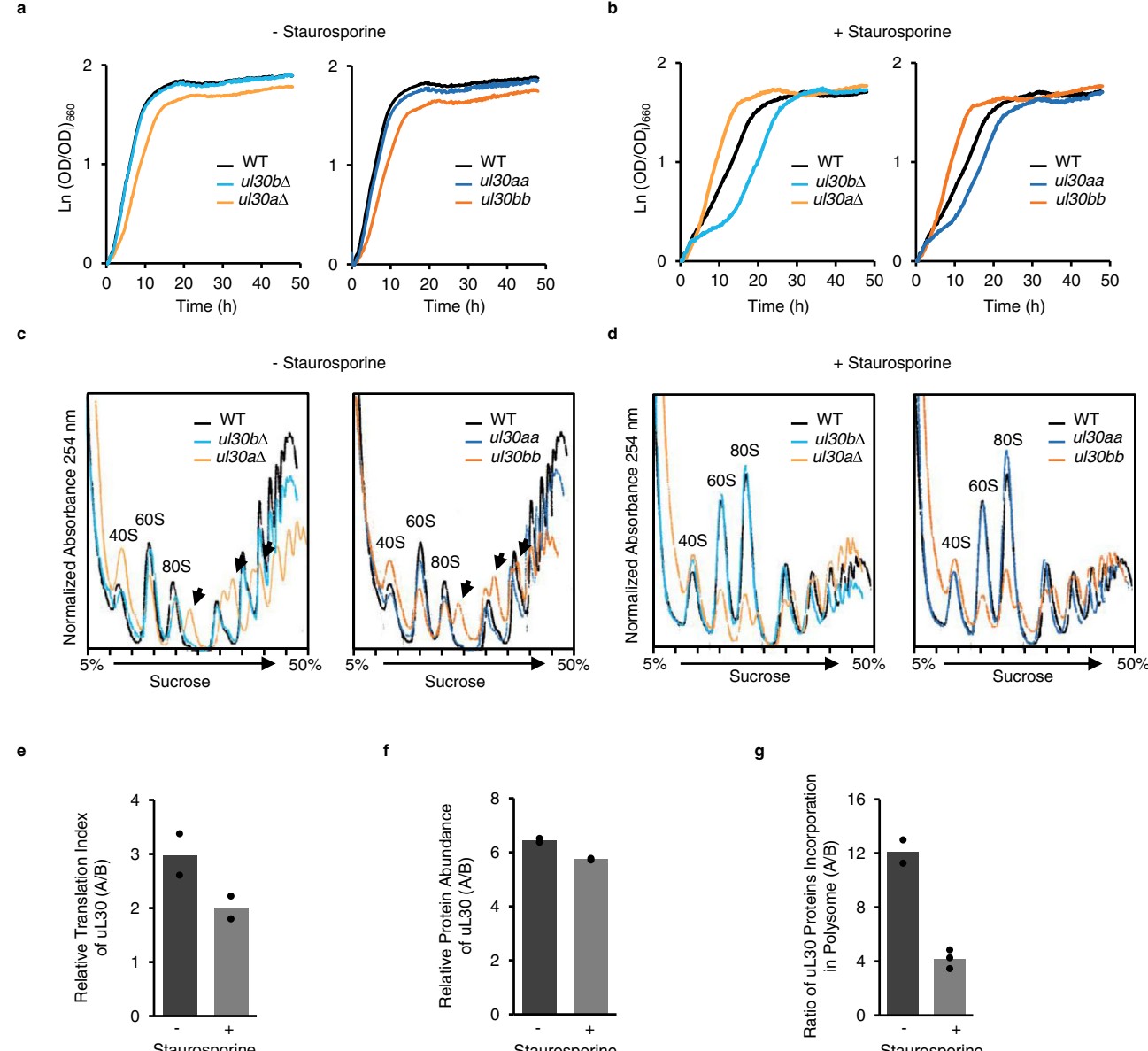

**Fig. 2 | *uL30* paralogs differentially alter cell growth, ribosome biogenesis and staurosporine resistance. a**, **b** Growth curves of wild-type (WT), deletion strains (*ul30bΔ* and *ul30aΔ*) and homogenized strains (*ul30aa* and *ul30bb*) grown in complete synthetic media without (**a**) or in the presence of staurosporine (**b**). **c**, **d** Polysome profiles were obtained from WT, cells lacking one paralog or expressing two copies of the same paralog from cultures grown in complete synthetic media without (**c**) or in the presence of staurosporine (**d**). Position of 40S and 60S ribosomal subunits and 80S ribosomes are indicated. Arrows indicate position of half-mers or 40S awaiting the 60S subunit. Curves shown in (**a**–**d**) are representative examples from *n* = 3 biologically independent biological samples. **e** Translation index (mRNA associated with polyribosomes/mRNA associated with monosome and subunits) was determined using RNA sequencing for each *uL30* paralog in absence (−) or presence (+) of staurosporine and the ratio for uL30A over uL30B is reported for *n* = 2 biologically independent biological samples. **f** Protein abundance of uL30 was determined by MRM in absence or presence of staurosporine and the ratio of uL30A over uL30B is reported for *n* = 2 biologically independent biological samples. **g** The amount of uL30 proteins incorporated into ribosomes was determined using MRM in absence or presence of staurosporine and the ratio of uL30A over uL30B is reported. Graphs in (**e**, **f**) show points from *n* = 2, while (**g**) show points for *n* = 2 (− staurosporine) and *n* = 3 (+ staurosporine) biologically independent biological samples.

known to reduce overall translation (Supplementary Fig. 4b). However, resistance to staurosporine was observed after the deletion of the gene coding for the translation elongation factor Tef4 suggesting that the paralog-specific effect of uL30 genes may stem from differences in their capacity to support translation elongation. The requirement of the minor paralog for staurosporine resistance is also supported by its increased expression in the presence of staurosporine. Exposing cells to staurosporine altered the ratio of uL30 proteins by favouring the translation of the minor paralog and its incorporation into active ribosomes (Fig. 2e–g and Supplementary Fig. 5). This suggests that the translation of uL30 paralogs is differentially regulated and as such we looked for features in this gene that could explain differences in translation. We did not find any significant difference in ORF size or UTR length but noted lower Kozak score and the clustering of sub-optimal codons at the 5′ end of *uL30B*. This may explain the preferential translation of uL30A under normal condition and the resistance of uL30B to staurosporine-dependent repression of translation (Supplementary Fig. 6). We conclude that the expression *of uL30A* is required for ribosome biogenesis while uL30B is needed to optimize the cell tolerance to staurosporine.

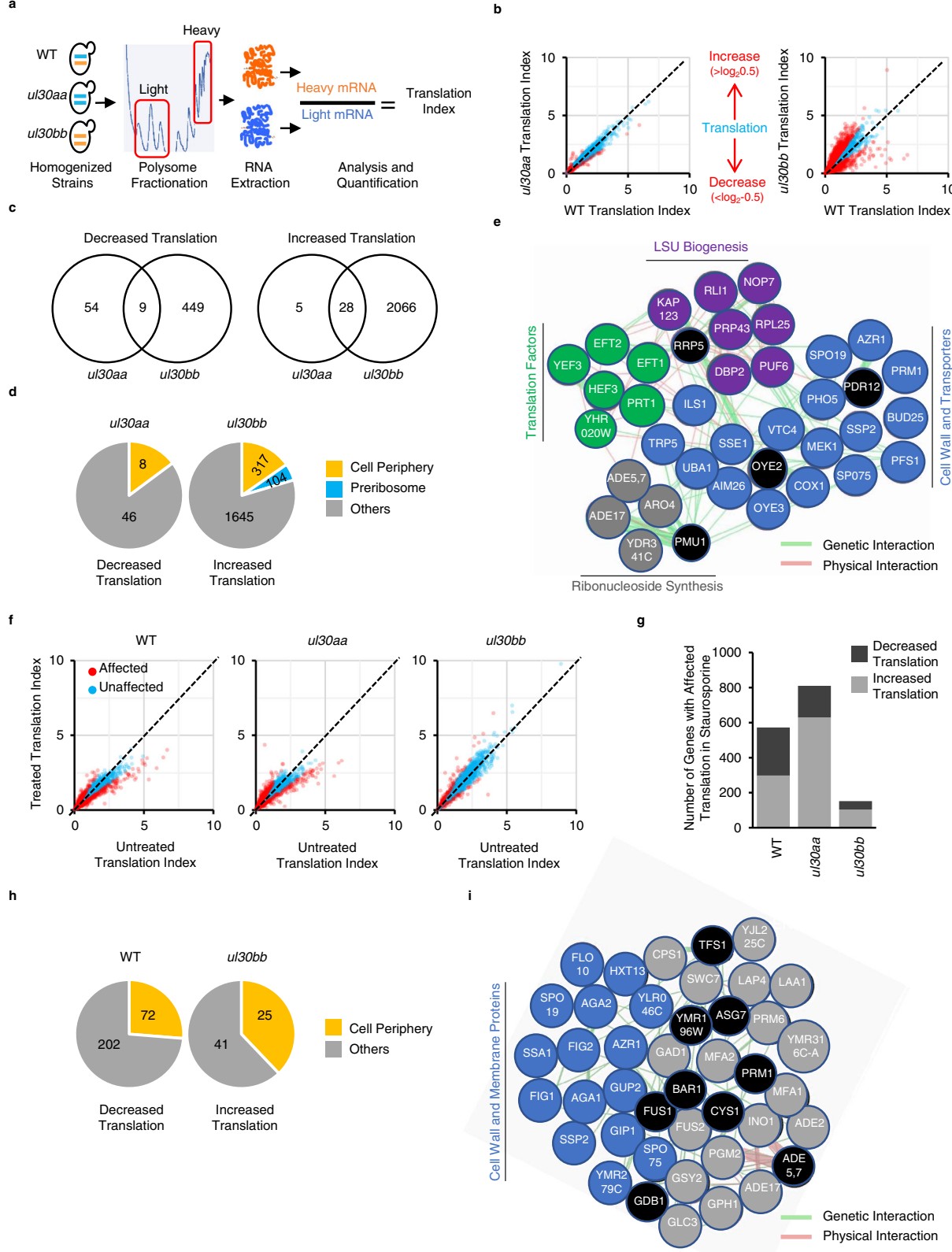

## The two paralogs of *uL30* modulate the translation of different subsets of mRNA

By comparing the translation pattern of cells expressing two copies of *uL30A or uL30B*, we identified a set of paralog-dependent genes. We sequenced ribosome-associated mRNAs, in wild-type, *ul30aa* and *ul30bb* cells, both before and after staurosporine treatment, and

verified the results by RT-qPCR (Fig. 3a and Supplementary Fig. 7a). We calculated the translation index as the ratio of mRNA associated with heavy (4 or more ribosomes) and light (subunits and monosome) ribosome fractions and verified the correlation of this presumed translation index or ribosome association ratio and the amount of protein produced by corresponding mRNA (Fig. 3a and

**Fig. 3 | Exposure to staurosporine modulates the translation of genes coding for cell periphery proteins in a paralog-dependent manner. a** Strategy for determining translation profiles of homogenized *uL30* strains. mRNA was extracted from heavy (polyribosome) and light (monosome and subunits) fractions and sequenced. Translation index was calculated as the ratio of mRNA associated with heavy to light fractions. **b** Translation index of mRNAs (expression >1 TPM) in homogenized strains is compared to that of wild-type. mRNAs showing differential translation by more than log₂ 0.5 difference are shown in red. **c** Venn diagrams of number of genes with changed association to polysomes comparing homogenized strains (*ul30aa* and *ul30bb*) to WT identified in (**b**). **d** Distribution of the number of genes present in enriched component gene ontology categories ($p < 0.001$ with Bonferroni correction) for genes with specific changes in association to polysomes. **e** Map of the genetic and physical interactions of the top up-translated genes in *ul30bb* strain was generated with the Genemania network construction tool. Up-translated genes were grouped and colored by functional pathways, genes in black

circles are not affected by *ul30bb*. **f** Translation index of mRNAs (expression >1 TPM) in WT and homogenized strains is compared in cells grown in absence (untreated) or in presence of staurosporine (treated). mRNAs showing differential translation by more than log₂ 0.5 difference are shown in red. **g** Bar graph showing the number of genes under-translated (dark gray) or over-translated (light gray) in response to staurosporine. **h** Distribution of the number of genes present in enriched component gene ontology categories ($p < 0.001$ with Bonferroni correction) for genes with specific changes in association to polysomes after staurosporine treatment. **i** Map of the genetic and physical interactions of the top up-translated genes in *ul30bb* strain in presence of staurosporine was generated with the Genemania network construction tool. Up-translated genes associated with cell wall and membrane component category were grouped and colored in blue, genes in gray associate with other categories and genes in black circles are not affected by *ul30bb*. RNA sequencing was performed on $n = 2$ biologically independent samples.

---

Supplementary Fig. 7b). Wild-type and *ul30aa* cells displayed similar polyribosome association patterns with only 96 genes more- or less-associated with polyribosome in *ul30aa* cells, suggesting that the minor paralog is required for the translation of a small subset of genes (Fig. 3b, c). In contrast, 2094 genes were more-associated with polyribosome and 458 less-associated with polyribosome in *ul30bb*, than in wild-type cells (Fig. 3b, c). This paralog-dependent change in global translation did not directionally correlate with changes in RNA abundance (Supplementary Fig. 8). Therefore, the paralog-specific changes in cell growth and drug resistance are likely related to changes in translation of a defined subset of genes.

Notably, the paralog-dependent changes in translation were not strictly linked to defects in ribosome biogenesis, or reduced ribosome numbers, since the paralog-specific effects on translation were also observed in ul30aa cells, which have similar ribosome biogenesis and ribosome number to those observed in wild-type cells (Fig. 2c and Supplementary Fig. 2f, g). Importantly we noticed that a set of 16 genes were inversely regulated (opposite effects observed in ul30aa and ul30bb cells) by *uL30* paralogs, underlining the paralog-specific effect on translation (Supplementary Data 5). Together these data indicate that uL30 paralogs differentially alter translation.

### The minor paralog of *uL30* induces the translation of cell periphery genes

Gene ontology analysis identified 'cell periphery' as the only enriched category that is inversely regulated by *uL30* paralogs. The expression of the A form decreases the translation of cell periphery genes (gene ontology enrichment with *p* value 6.3E−4), while the B form increases the translation of genes in this category (*p* value 1.0E−7; Fig. 3d). Since most cell periphery genes are translated in the endoplasmic reticulum (ER) we examined the effect of uL30 paralog on the translation of ER-translated genes[23]. Interestingly, we also found that ER-translated genes[23] are over-translated in our *ul30bb* strain with two copies of uL30B (Supplementary Fig. 9). This further supports the link between uL30 paralog and translation of cell periphery genes and is consistent with the proximity of uL30 and its N-terminal domain to the peptide exit channel (Supplementary Fig. 2a). Analysis of the genetic and physical interactions of the top 35 translated genes in *ul30bb* cells identified a tight network of proteins involved in translation, large subunit (LSU) biogenesis, ribonucleoside synthesis, cell wall and transport (Fig. 3e). The paralog-dependent changes in the translation of this network of related genes may explain the paralog-dependent tolerance to staurosporine treatment. Staurosporine affects cell wall integrity through the inhibition of protein kinase C (PKC) and as such the paralog-dependent changes in the expression of cell wall and membrane proteins may alter the drug effect on cell growth[24,25]. We conclude that *uL30* paralogs differentially modulate the translation of cell periphery genes, and that this likely explains the paralog-dependent differences in staurosporine resistance.

### The minor paralog of uL30 tempers staurosporine-dependent modification of translation

Staurosporine altered the translation of ~600 genes in wild type and this number of staurosporine-dependent genes was increased in *ul30aa* cells to 800 and decreased in *ul30bb* cells to only 150 genes (Fig. 3f, g). This indicates that the effect of staurosporine on translation is modulated in a paralog-dependent manner. This paralog-dependent modification of the translation response to staurosporine, at least in the case of ul30aa, occurs even when ribosome biogenesis and ribosome numbers are identical (Supplementary Fig. 2g). Interestingly, the only set of genes that were inhibited in wild-type and induced in *ul30bb* cells after staurosporine treatment were those associated with cell periphery (Fig. 3h). Genes that were most upregulated by staurosporine in the presence of the B, and not the A form were also associated with the cell wall and membrane proteins (Fig. 3i).

These data indicated that *uL30B* increases resistance to staurosporine by increasing the translation of cell periphery genes. We therefore hypothesized that the minor paralog of *uL30* might affect the response to cell wall integrity drugs other than just staurosporine. As postulated, cells expressing the A form were sensitive, while those expressing the B version were resistant to the cell wall integrity drugs ketoconazole and caffeine (Supplementary Fig. 10). We conclude that *ul30bb* dependent resistance to drugs is not limited to staurosporine but extends to other drugs affecting cell wall integrity.

### uL30 minor paralog selectively induces the translation of mRNAs with long open reading frames

Examining the features of the genes that are differentially translated by *uL30* paralogs identified clear differences in the length of the open reading frames. Coding regions of the top 20 genes, which are more translated in the presence of *uL30b*, are on average >3 times longer than those that are under-translated (Fig. 4a). Comparison between the average ORF length of all the genes that are translated in a paralog-dependent manner with that of all genes in yeast indicated that in general cells expressing uL30B differentially translate long genes over the short genes (Fig. 4b). This change in translation was not linked to changes in mRNA abundance since we did not see any correlation between ORF length and RNA abundance in ul30aa and ul30bb cells (Supplementary Fig. 8c). The paralog-dependent change in translation of long ORFs is not linked to translation efficiency. Most of the long genes that are differentially translated between ul30bb and ul30aa cells have average translation index in wild-type cells (when calculated as the ratio of mRNA associated with heavy polyribosome/mRNA associated with light ribosomes (Supplementary Figs. 11a and 12). However, we found that the ribosome density decreases as the ORF length increases as previously observed (Supplementary Fig. 11b)[26]. Therefore, the increased translation of long ORFs in *ul30bb* cells is not linked to the fraction of the mRNA associated with heavy polysomes in wild type but could be linked to ribosome density. Remarkably,

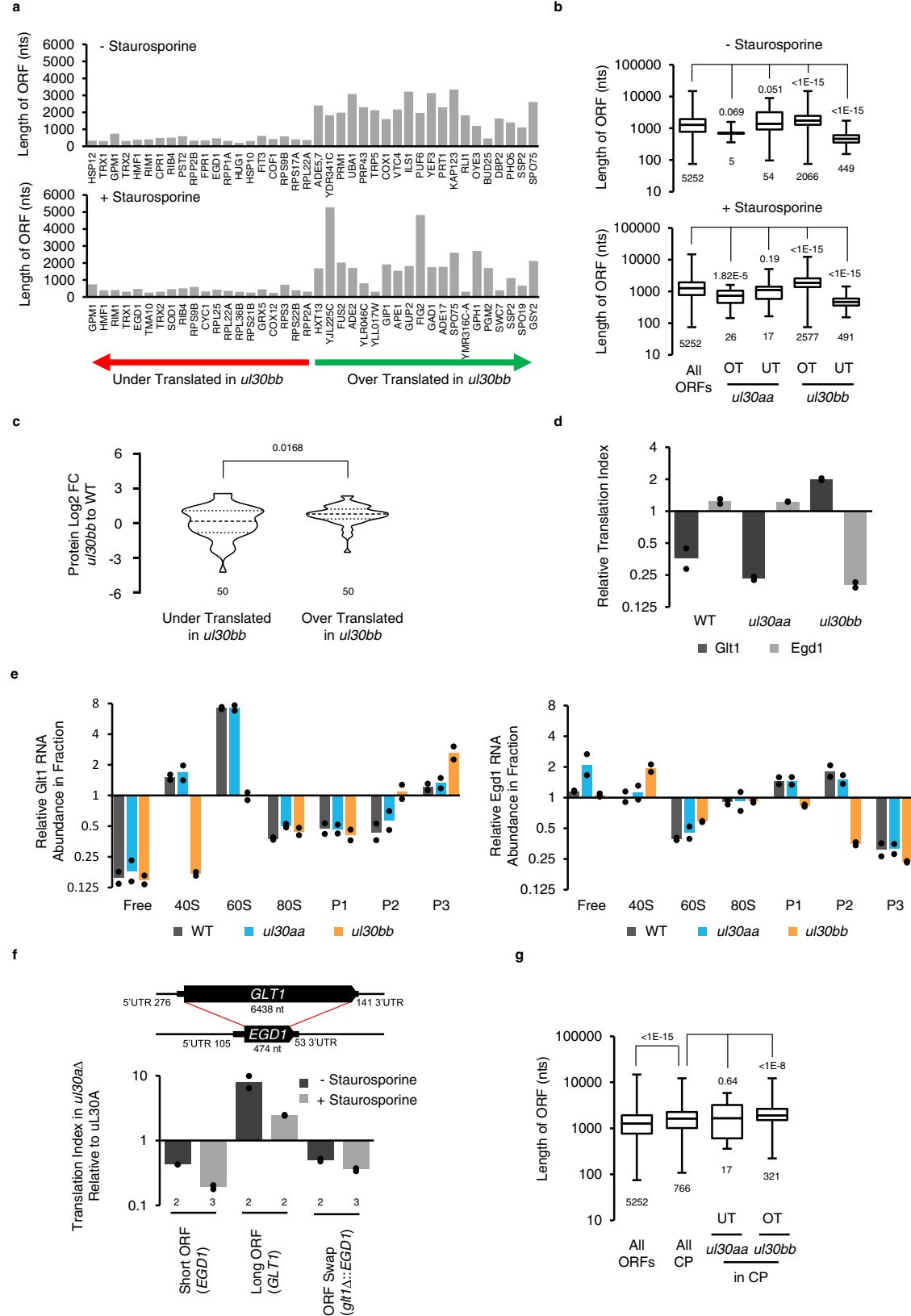

expression of long and short ORFs were inversely affected in *ul30aa* and *ul30bb* cells upon exposure to staurosporine. For example, while ORFs over-translated in response to staurosporine in *ul30aa* were relatively short, those over-translated in *ul30bb* were long when compared with wild-type cells (Fig. 4b, lower panel). Therefore, the observed ORF length-dependent change in translation after

staurosporine is not restricted to *ul30bb* cells, which exhibit difference in ribosome biogenesis, but instead extends to *ul30aa*, which displays no major defect in ribosome production (Supplementary Fig. 2f, g). The effect of the shift in mRNA association to ribosomes on protein abundance was examined by mass spectrometry. As indicated in Fig. 4c, the protein abundance in *ul30bb* was reduced for genes

**Fig. 4 | Paralogs of *uL30* modulate cell response to drug by altering the translation of long genes. a** Top 20 genes over- or under-translated in *ul30bb* in absence (top panel) or presence (lower panel) of staurosporine are plotted relative to ORF length in nucleotides. Genes are ordered on the X-axis according to the magnitude of change in translation. **b** Box plots comparing the ORFs length of all genes and genes that are over-(OT) and under-translated (UT) in *ul30aa* and *ul30bb* in absence (top panel) or presence (lower panel) of staurosporine. Box limits represent the first and third quartiles, the middle line is the median and whiskers extend to minima and maxima. ORFs numbers in each category is stated at bottom. The results of two-tailed Mann–Withney test indicated on top. **c** The violin plots show the relative change in the protein abundance of the top 50 OT and UT genes in *ul30bb*. The data obtained from *n* = 4 (WT) and *n* = 2 (*ul30bb*) biologically independent samples. The result of two-tailed Mann–Withney test indicated on top. **d** Translation index of long (Glt1, 6438 nt) and short (Egd1, 474 nt) ORFs

determined by RT-qPCR and normalized to middle size ORF (Osh6, 1344 nt). The black circles indicate data from *n* = 2 biologically independent samples. **e** Relative enrichment of Glt1 and Egd1 was monitored across sucrose gradients fractions as described in (**d**). P1 are di-trisomes, P2 are quadra-penta-hexasomes and P3 are heptasomes to decasomes fractions. **f** *GLT1* ORF was substituted by *EGD1* ORF (top panel) in strains containing or lacking major paralog of uL30. Graph shows the relative change in translation for the endogenous *EGD1*, *GLT1*, and the chimeric gene (*glt1Δ::EGD1*) upon deletion of uL30A in absence (−) or presence (+) of staurosporine. The number of independent biological replicates is indicated at bottom. **g** ORFs length of cell periphery proteins (CP) genes and genes that are under- (UT) or over-translated (OT) in *ul30aa* and *ul30bb* strains are represented as in (**b**). The RNA sequencing data in (**a**, **b**, **g**) are generated using *n* = 2 biologically independent samples.

showing decreased association with ribosome and increased for those showing increased association with ribosome. This confirms that in most cases the change in mRNA association with ribosome associates with changes in the amount of proteins produced in the cell.

The ORF length-dependent translation was next validated by RT-qPCR using the long *GLT1* (6438 nt) and short *EGD1* (474 nt) model ORFs (Fig. 4d). Once again, the translation of long and short ORFs were inversely regulated in *ul30aa* and *ul30bb* cells. The differences in the translation of mRNAs of different lengths was not due to differences in the early initiation phases, as the amount of free mRNA did not vary in cells expressing different paralogs (Fig. 4e). Instead, we observed increased association of the long ORFs with the heavy polyribosome in cells expressing uL30B. Curiously, this minor isoform also decreased the ratio of long/short ORF co-sedimentation with the 40S and 60S fractions, mostly due to an increased association of the short ORF mRNA in these fractions (Fig. 4e). It is not clear if this unusual co-sedimentation pattern with the 60S fraction represents an abnormal association with the subunit or a coincidental co-sedimentation of an independent mRNA protein complex. In all cases, it is clear that uL30B favors the optimal translation of long ORFs, through increased association with ribosomes.

To further investigate the importance of ORF length to uL30B-dependent translation, we replaced *GLT1*'s long ORF with the short *EGD1* ORF and monitored translation before and after staurosporine treatment. Strikingly, changing the ORF length abolished both paralog- and staurosporine-dependent translation, confirming the importance of ORF length to uL30B effect on translation (Fig.4f and Supplementary Fig. 13). Interestingly, ORFs coding for cell periphery proteins, as a group, are longer than the average ORF length, explaining their increased expression in the presence of uL30B (Fig. 4g). We conclude that the *uL30* paralog-dependent response to staurosporine is mediated by differential translation of genes with long ORFs.

## uL30 minor paralog-dependent drug resistance is not due to differences in ribosome biogenesis
To evaluate the contribution of their chromosomal loci and regulatory sequences to the paralogs' functional specificity, we compared cells expressing the cDNA of either paralog from identical plasmids and promoters (Fig. 5a, b). The cDNAs were transcribed from the promoter of the 40S subunit protein *eS28A/RPS28A* as the sole source of uL30 protein in the cell. Unlike the native copy, the plasmids borne copies of uL30A or uL30B equally increased the abundance of total uL30 mRNA produced in the cell (Fig. 5c and Supplementary Fig. 14a). As would be expected, the increased mRNA abundance did not increase the amount of uL30 proteins, as free RPs are rapidly degraded (Supplementary Fig. 14b)[27]. Importantly, plasmids did not alter the amount of other RPs or rRNA suggesting that the ribosome numbers in these cells remain similar (Supplementary Fig. 14c–e). However, expression of the ORFs from the plasmid eliminated differences in the paralogs' Kozak scores and thus reduced the preferential association of the A mRNA

with heavy polyribosomes (Supplementary Fig. 6). The lower Kozak scores and reduced association with heavy polyribosome of uL30B that favors the translation of uL30A are altered when it is expressed from plasmid. This indicates that the native and plasmid-borne copies of the uL30 are not translated in the same way. Together these results indicate that cells expressing *uL30* copies from plasmids generate equal amount of protein forms and ribosomes but differ from wild-type cells in paralog mRNA abundance and translation pattern.

Comparison between cells expressing different versions of *uL30* genes from plasmids indicated, that unlike the chromosomal copies, the plasmid-borne versions equally supported growth and subunits production under normal conditions (Fig. 5d–f). This might be due to the difference in expression and translation of the plasmid-borne copies. However, despite the lack of differences in ribosome biogenesis, cells expressing the B form from the plasmid were more resistant to staurosporine than those expressing the A form (Fig. 5d). The B form-specific increase in translation of long ORFs and cell periphery genes was also maintained in cells expressing the paralogs from plasmids (Fig. 5g–i). The differential effects of the uL30 paralog on the translation of long ORFs was also maintained when they are expressed from plasmid and was confirmed using RT-qPCR (Fig. 5j, k and Supplementary Figs. 14f, 15). This indicates that uL30 paralog may differentially alter the translation pattern independent of any regulatory or intervening sequence (e.g., introns or snoRNA) or significant variations in ribosome production or number (e.g., the 60S biogenesis defects). We conclude that the uL30 minor paralog-specific translation and drug resistance phenotypes do not depend on defects in the synthesis of the 60S subunit.

## The different functions of the *uL30* paralogs are driven by differentially acetylated N-terminal domain sequences
Most of the differences between the uL30 copies are found in the first 42 amino acids of the N-terminal domain (Fig. 6a and Supplementary Fig. 6b). Indeed, the N-terminus includes 4 out of the 5 non-identical amino acid that distinguish between the uL30 versions and features paralog-specific acetylation. In total extract, the N-terminal serine residue of the B form is 100% acetylated, while the alanine residue at the corresponding position of the A version is mostly unmodified (Fig. 6a, b). The acetylation level of the A, and not the B, form depends on the levels and ratios of the two paralogs. In strains containing plasmids encoding the uL30A paralog under the control of a heterologous promoter the percent of acetylated A form was 75%, compared to 10% in wild type. In contrast, expression from a plasmid had no effect on the mostly acetylated B form (Supplementary Fig. 16c). This lack of difference in B form acetylation may explain the reduced difference between the A and B phenotypes when the paralog is expressed from plasmids.

The acetylated uL30 proteins were more abundant in heavy polyribosomes, and exposure to staurosporine increased the amount of ribosome incorporating the 100% acetylated B form (Figs. 6c, 2g and Supplementary Fig. 16). Notably, the small proportion of uL30A proteins that were acetylated were also preferably found in heavy

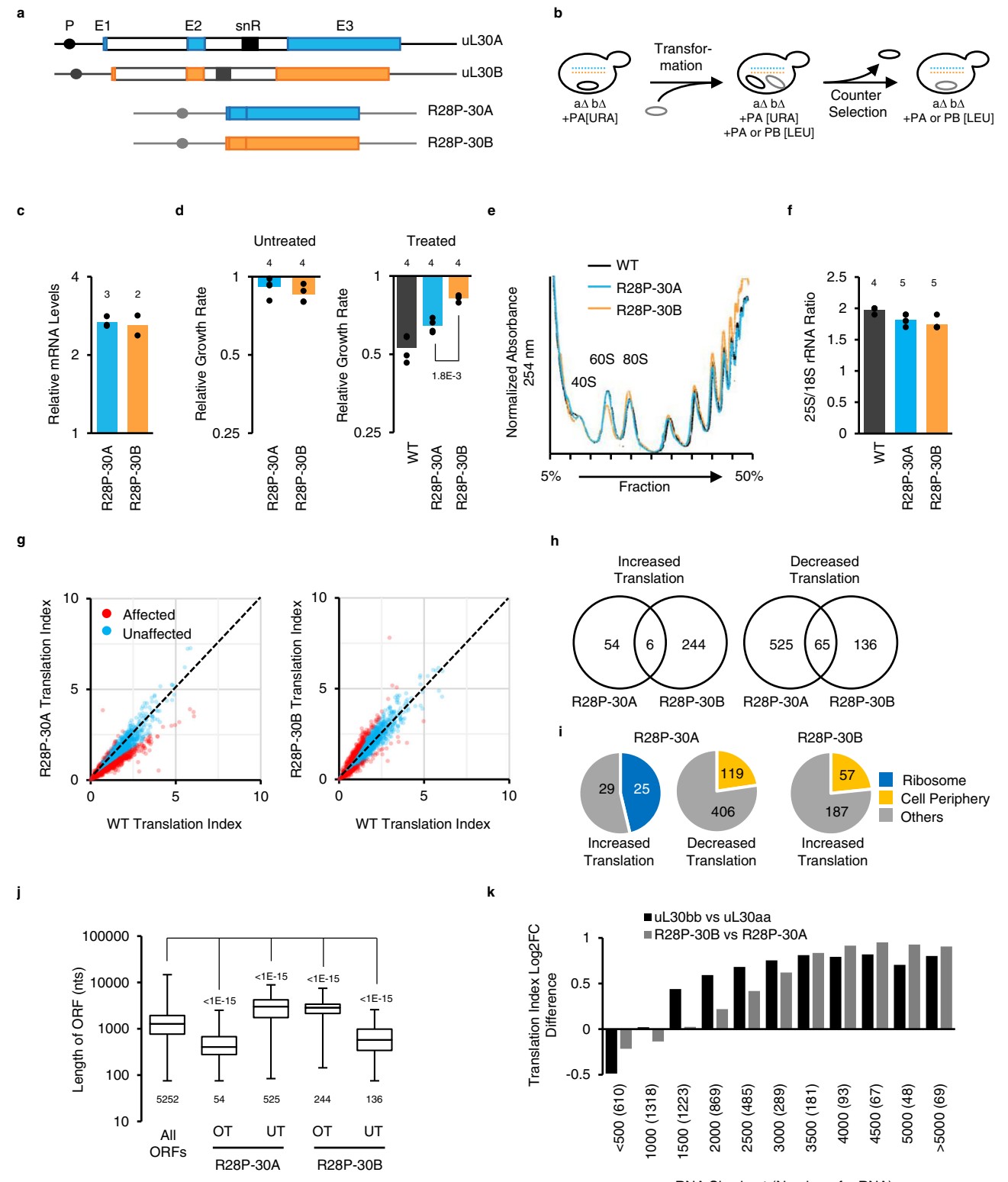

polyribosome (Supplementary Fig. 16a). This indicates that ribosomes with acetylated uL30 are more likely to be in heavy polyribosomes, which normally are associated to longer ORFs. We conclude that the minor paralog of uL30 is differentially acetylated and incorporated into actively translating ribosome.

The differential acetylation of uL30 paralogs and their resulting effects on translation and staurosporine resistance depend on the sequence of the N-terminal domain. As indicated in Fig. 6d, swapping the first 42 amino acids of the chromosomal copy of the B form with

the A version abolished the hyper-acetylation of the B form. As expected, this swap (uL30B-Ant), which only modifies the amino acid sequence of the uL30B N-terminus, without affecting the overall chromosomal structure of either A or B genes or ribosome numbers, also increased cell sensitivity to staurosporine and reduced the translation of long ORFs (Fig. 6e–h).

Next, we swapped the N-terminal domains of uL30A and uL30B in the context of the same plasmid and promoter sequences (Supplementary Fig. 17a). Swapping the N-terminus on a plasmid thus

**Fig. 5 | *uL30B* promotes translation of long mRNAs and staurosporine resistance independent of the noncoding regulatory sequences. a** Endogenous wild-type genes are shown on top and those of plasmids expressing paralogs cDNA from *RPS28A* promoter and *ADH1* terminator sequences are below. Blue, orange and white boxes indicate exons of *uL30A*, *uL30B* and introns. Lines show 5' and 3' UTRs. P, E1, E2, E3 and snR indicate position of promoter, exon 1–3, and snoRNA. **b** Strategy for creating haploid yeast strains expressing a single copy of uL30 genes from plasmids. **c** uL30 mRNA expression from plasmid was quantified by RT-qPCR relative to endogenous mRNA. **d** Growth of cells harboring R28P-30A or R28P-30B plasmids compared to that of WT in absence (left panel) or presence (right panel) of staurosporine. The results of two-tailed paired *t* test indicated at bottom. **e** Polysome profiles of cells expressing uL30 from plasmid. Curves are representative examples from *n* = 3 biologically independent samples. **f** The ratio of 25S to 18S rRNA was determined by capillary electrophoresis. **g** Translation index of mRNAs (expression > 1 TPM) detected in R28P-30A or R28P-30B was compared to that of wild-type. Differentially translated mRNAs (difference > log₂ 0.5) are shown in red. **h** Venn diagrams of the number of genes showing differential polyribosome association in (**g**). **i** Number of differentially translated genes in each enriched component gene ontology category (*p* < 0.001 with Bonferroni correction). **j** Box plots representing the length of differentially translated ORFs. Box limits represent the first and third quartiles, the middle line is the median and whiskers extend to minima and maxima, the number of ORFs in each category is stated under the boxes, the results of two-tailed Mann–Withney test are stated above. **k** Bar graph showing the relative translation index for B/A paralogs according to mRNA sizes. RNA sequencing for (**g**–**k**) was performed on *n* = 2 biologically independent samples. The number of biologically independent samples shown in (**c**, **d**, **f**) is stated above the bars representing the means.

produced the chimeric protein as the sole source of uL30. This clearly also swapped the effect of the paralogs on growth, staurosporine resistance and translation of long ORFs, with little effect on ribosome numbers, biogenesis, or overall polyribosome profile (Supplementary Fig. 17). The lack of connection between the paralog-specific translation and tolerance to staurosporine is also clear form the general comparison of the rRNA and growth rate of the different strains used in this study (Supplementary Figs. 18 and 19). All strains except those showing uL30A dependent defect in ribosome biogenesis (*ul30aΔ* and *ul30bb*) displayed growth rates and rRNA amount similar to that detected in wild-type cells. Exposure to staurosporine also had no effect on ribosome number in any of the strains tested compared to wild type (Supplementary Fig. 19a, b). Interestingly, we found that exposure to staurosporine induces the expression of uL30B when uL30A is absent (*ul30aΔ and ul30bb*) and represses the expression of uL30A when uL30B is absent (*ul30bΔ*) (Supplementary Fig. 19c). This is consistent with the presumed cells' preference for expressing uL30B when exposed to staurosporine. We conclude that the N-terminal domain of the uL30 paralogs differentially affects cells tolerance to staurosporine.

## Discussion

In this study, we show that exposure to drugs modifies ribosome composition and translation by altering the ratio of proteins generated from duplicated ribosomal protein genes. Most yeast ribosomes are generated from one predominant (major) housekeeping paralog required for normal growth and a minor paralog needed under certain conditions (Fig. 1). Interestingly, we discovered that quasi-identical RPGs could produce two differentially modified proteins with different effects on translation. For the uL30 gene pair the major paralog is hypo-acetylated and required for optimal ribosome biogenesis and cell growth, while the minor form is hyper-acetylated to provide optimal drug resistance (Figs. 2, 4 and 6). Paralog-dependent differences in drug-resistance originate from variation in the sequence of their differentially acetylated N-terminal domains (Fig. 6). Together our data provide clear evidence of programmed growth condition-specific changes in ribosome composition that promote drug resistance by modifying translation.

The paralog-specific effects of *uL30* include differences in 60S biogenesis, translation and staurosporine resistance. The different effects of the paralogs on ribosome biogenesis are consistent with the previously established role of *uL30* in assembling the 25S rRNA domains[28]. The synthesis of the 60S subunit is dependent on the expression of the chromosomal copy of the major paralog and its function cannot be restored by the expression of two copies of the minor isoform *uL30B* (Fig. 1). The major paralog function in ribosome biogenesis is not linked to the intron, or the snoRNA embedded therein, since intron deletions of either paralog did not affect ribosome production (Fig. 5e)[29]. Instead, uL30B protein appears to be less efficient than uL30A in supporting ribosome biogenesis, a defect that could be overcome by increasing the expression of uL30B from plasmids (Fig. 5). Expression of uL30 genes from plasmids modifies the paralog mRNA amounts, translation pattern and post-translation modifications, which may in turn alter the proteins' natural functions and localization and consequently blur the difference in their capacity to support ribosome biogenesis (Supplementary Figs. 6, 14 and 16)[30]. In all cases, the difference in the capacity of uL30 paralog to support ribosome biogenesis is not strictly linked to the amount of mRNA produced by these genes but the location from which they are expressed as demonstrated by the comparison between the *ul30aa* and *ul30bb* cells (Fig. 2 and Supplementary Figs. 2 and 17).

In contrast to ribosome biogenesis, the copy-specific effect of uL30, on translation and resistance to drugs, cannot be erased by the expression of the paralog from plasmid (Fig. 5). Instead, the copy-specific effect of uL30 depends at least in part on the sequence of the N-terminal domain, which features paralog-specific structure and acetylation patterns (Fig. 6 and Supplementary Fig. 2). The effect on translation and drug resistance is not strictly linked to defects in ribosome synthesis or ribosome number. Indeed, paralog-specific differences in translation and tolerance to staurosporine are not restricted to cells exhibiting defects in ribosome biogenesis (e.g., *ul30aΔ* and *ul30bb*) but extends to cells with no observable defects in ribosome production (e.g., *ul30aa*, uL30B-Ant, R28P-30A, and R28P-30B, Figs. 2, 6 and Supplementary Figs. 2, 17).

The effect on translation is not due to pleiotropic reprogramming of translation, like previously suggested in the case of the knockout of the mammalian eS25 genes[31]. The effects of the paralog deletion could be completely restored by expressing a copy of the native gene from plasmid. In addition, we can see the difference in translation in identical yeast strains transformed by plasmids that differ only in the amino acid sequence of uL30 (Supplementary Fig. 17). Most importantly the paralog-specific effects were not linked to a single deletion or mutation but could be reproduced in multiple strains and by mutations in both plasmid and chromosome borne genes (Figs. 2, 5 and 6).

The data presented here support earlier work suggesting that gene duplication in yeast permits the cell to express one major copy for housekeeping functions and another for special growth conditions or stress[18,22,32]. In the case of uL30, it is the A form that is better expressed and translated under normal condition, while the B form becomes more expressed after exposure to drugs. The increased abundance of the B form under stress is achieved through the decreased translation of its major counterpart. This could be explained by the general tendency of RNAs with less optimal codons and a weaker Kozak scores to be more resistant to stress than highly translated transcripts[33,34]. Indeed, we have previously found that in general the major copies of duplicated ribosomal protein genes are more likely to be inhibited under stress than the minor copies[22,29]. In theory, differences in the paralogs' responses to stress could also be achieved by specific stress response elements but we have not found any obvious candidates so far.

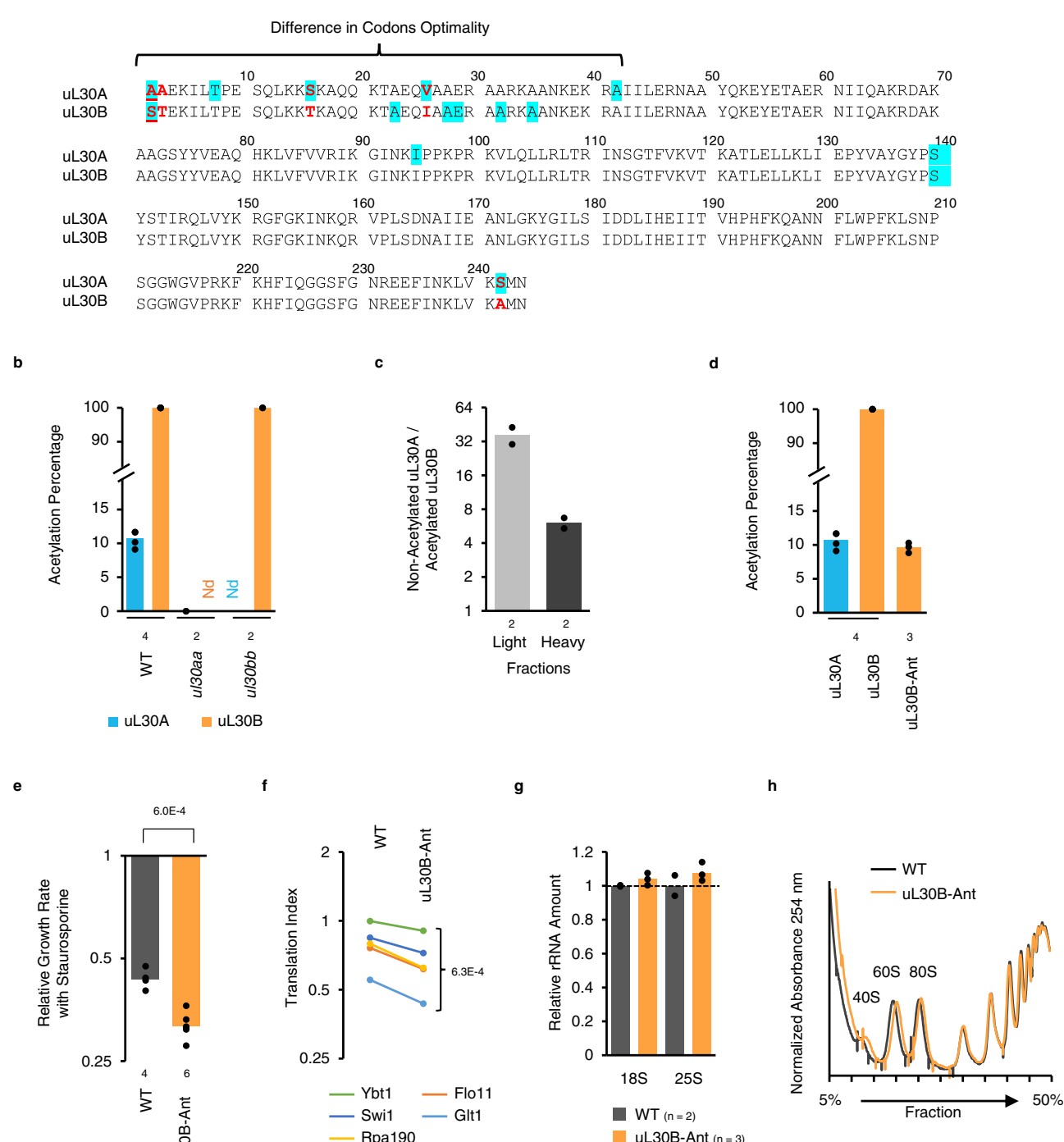

**Fig. 6 | Post-translation modification alters function of uL30 paralogs. a** Amino acid sequences of uL30 paralogs were aligned using Clustal/W2. Different amino acids are indicated in red, amino acids encoded by suboptimal codon are highlighted in cyan and position of N-terminal acetylation is underlined. **b** N-terminal acetylation was measured by mass spectrometry in wild-type, *ul30aa* and *ul30bb* cells. The number of biologically independent samples is indicated at bottom. **c** N-terminal acetylation level of uL30 proteins detected in light (60S) and heavy (heptamers and heavier) polyribosome fractions in WT cells using mass spectrometry. The number of biologically independent samples is indicated at bottom. The ratio of non-acetylated uL30A over acetylated uL30B in each fraction is reported. **d** N-terminal acetylation was measured by mass spectrometry in wild-type and uL30B-Ant mutant. The number of biologically independent samples is indicated at bottom. **e** Relative growth rates of WT and uL30B-Ant mutant after exposure to staurosporine. The results of the two-tailed *t* test assuming unequal variance are indicated on top. The number of biologically independent samples is indicated at bottom. **f** Translation index of 5 long mRNAs (3945-6438 nt) that were found down regulated in *ul30aa* by RNA sequencing was analyzed by RT-qPCR in WT and uL30B-Ant. The result of the paired two-tailed *t* test of the change in the translation of the indicated genes is shown on the right. **g** 18S and 25S rRNA was quantified by capillary electrophoresis and reported relative to WT. The number of biologically independent samples is indicated at bottom. **h** Polysome profiles of WT and uL30B-Ant. Curves are representative examples of biologically independent samples *n* = 2 WT and *n* = 4 uL30B-Ant.

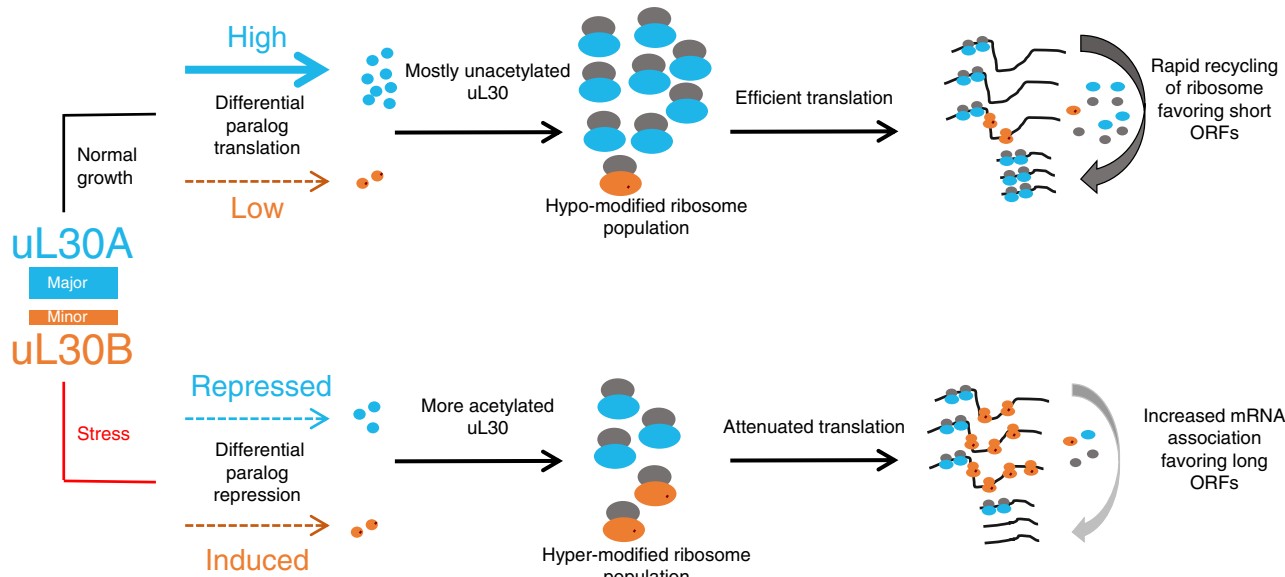

**Fig. 7 | Schematic representation of a hypothetical model for paralog-dependent modulation of translation.** Expression of the major and minor ul30 paralogs and their impact on translation is depicted under normal growth conditions or after exposure to stress (e.g., staurosporine). In this model, the major paralog (uL30A, shown in blue) is predominantly translated under normal growth condition (shown on top) leading to the generation of a highly active population of ribosomes incorporating the hypo-modified version of uL30. These ribosomes support efficient translation cycles and reduce the association time with long ORFs which in turn permits repeated translation of genes with short ORFs. In contrast, when cells are exposed to drugs (shown at bottom), the translation of the major paralog is inhibited thus increasing production of a less active population of ribosomes incorporating the hyper-acetylated minor paralog (uL30B, shown in orange). This change in the ribosome profile slows translation resulting in an increase in the number of ribosomes associated with long ORFs. This reduces the number of ribosome free to reinitiate subsequent translation rounds and by consequences decrease the translation of short ORFs. Large ribosomal subunits incorporating major and minor paralogs are indicated as blue and orange ovals. Post-translational modification of the minor paralog is indicated as orange dots and mRNAs are indicated as black lines.

In this model, the abundance of hypo-acetylated uL30A form ribosome ensures optimal translation, especially of short mRNAs, which in turn ensures the most rapid elongation and recycling rate (Fig. 7). In contrast, translation is inhibited under stress, reducing translation of the major paralog, with little effect on the already under-translated minor version. This mechanism is supported by previous studies showing that under-translated genes are better translated under stress and that most major and not minor, paralogs are repressed under stress[33,34]. This differential inhibition of uL30 paralog increases the number of ribosomes containing the hyper-acetylated minor protein form, slowing-down translation further and increasing the association of ribosomes with long mRNAs. In this model then, the paralogs change the translation equilibrium of long and short ORFs. This switch is likely due to reduced availability of free 60 S subunits, either through reduced production of the 60S, as in the case of *ul30bb* cells (Fig. 2), or by delayed release from the mRNA, as would be the case in cell expressing a single copy of uL30B from plasmid (Fig. 5). This model is not necessarily inconsistent with earlier reports suggesting that ribosomal protein depletion or changes in ribosome numbers may alter translation[34]. Reducing ribosome activity by increasing the expression of a less active form (e.g., the minor copy) of ribosomal proteins may mimic the effect of ribosomal protein depletion without reducing the overall amount of ribosomal proteins or ribosome produced.

In summary our work shows that the duplication of ribosomal protein genes provides the cells not only with greater flexibility for fine tuning ribosomal protein expression but also may provide means for shifting the translation equilibrium under adverse or suboptimal growth conditions.

## Method
### Experimental model
*Saccharomyces cerevisiae* strains used in this study share the BY background[35] and are described in Supplementary Data 1. Cultures were grown in yeast complete (YC) media (1.7 g/l yeast nitrogen base without amino acids and ammonium sulfate, 1 g/l sodium glutamate, 100 mg/l cysteine and threonine, 85 mg/l tryptophan, 80 mg/l leucine, 60 mg/l lysine, 50 mg/l aspartic acid, isoleucine, methionine, phenylalanine, proline, serine, tyrosine, uracil and valine, 20 mg/l adenine, arginine and histidine, pH 6) or YEPD (5 g/l yeast extract, 10 g/l peptone, 100 mg/l adenine) with 2 g/l dextrose at 30 °C. Strains were constructed using standard procedures[36] and gene replacements verified by PCR using oligonucleotides described in Supplementary Data 2. Plasmids R28P-30A, R28P-30B, R28P-30A-Bnt, and R28P-30B-Ant were synthesized and sequenced by Biobasic.

### Growth assays
Overnight saturated starter cultures from freshly streaked single colonies were used to inoculate pre-cultures grown to reach a cell density of 1.67–3.33E7 cells/ml after an overnight incubation. These pre-cultures were diluted the next morning to 0.83–1.25E6 cells/ml and the experiment started when the cultures reached 2.33–2.67E6 cells/ml. For growth assays in tubes, IC 50 (drug concentration which reduces the growth of wild-type cells by 50%) quantities of drugs were added at this point and absorbance at 660 nm was measured at regular intervals for 10–48 h. For growth assays in 96-well microplates, the absorbance at 660 nm of 100 μl cultures was measured at 10 min intervals for 48 h in a Biotek Powerwave or Epoch2 readers. The maximal growth rates (doubling times) for each strain were calculated as previously described[37]. Doubling times of all uL30 strains are reported in Supplementary Data 4.

### Total RNA isolation
Cells from 5–10 ml cultures grown to 1–1.33E7 cells/ml were collected by centrifugation, washed with DEPC-treated water, and resuspended in 300 μl of LETS buffer (0.01 M Tris-HCl pH 7.5, 0.1 M LiCl, 0.01 M Na₂EDTA pH 8.0, 0.2% SDS), then 300 μl of phenol and 1 volume of

acid-washed glass beads were added. Cells were broken by 10 cycles of vortexing for 30 s and cooling on ice for 30 s. Proteins were removed by phenol:chloroform extraction and the RNA precipitated by the addition of 2.5 volumes of ethanol and 0.3 M sodium acetate pH 5.5. RNA samples were resuspended in 50 μl of DEPC-treated H$_2$O and quantified by Nanodrop. Samples diluted to 25–100 ng/μl were used for 25S/18S quantification on an Agilent TapeStation 2200 using RNA screen tape.

### Polysome and RNA preparation for translation index analysis

Polysomes were prepared as described by[38,39] with the following modifications. 50 ml pre-cultures made by inoculating with a single colony from a freshly streaked plate and incubated overnight to reach a cell density of 5.83–8.33E7 cells/ml. The pre-cultures were diluted to 0.83E6 cells/ml in a volume of 400 ml. Staurosporine treatment started when the cultures reached a density of 3.33E6 cells/ml and ended when the cultures reached a density of 1–1.33E7 cells/ml. The cells were collected in precooled 400 ml plastic bottles, cycloheximide was added to a final concentration of 0.1 mg/ml and they were immediately spun down at 3000 g for 5 min at 4 °C. The pellets were washed twice in cell lysis buffer (20 mM Tris-HCl pH 8, 140 mM KCl, 1.5 mM MgCl$_2$, 0.5 mM DTT, 0.1 mg/ml cycloheximide, 1% Triton X-100, 1 mg/ml heparin). The washed cells were resuspended in 3 ml of lysis buffer, 500 μl of acid-washed glass beads were added, and the samples were vortexed 4 × 20 s at full speed on a vortex Genie 2 (VWR Scientific Products). The cellular debris were removed by centrifugation at 2600 × $g$ for 5 min at 4 °C and 1.5 ml of the lysate was transferred to a microtube and centrifuged again at 9500 × $g$ for 5 min at 4 °C. Within 2.5 h, the supernatant was layered onto a 5–50% sucrose gradient prepared with 20 mM Tris-HCl pH 8, 140 mM KCl, 5 mM MgCl$_2$, 0.5 mM DTT, 0.1 mg/ml cycloheximide, 1 mg/ml heparin on Gradient Master 108 (Biocomp) at an angle of 60° for 5 min at 25 rpm and placed at 4 °C for at least 10 h) and centrifuged at 34,000 × $g$ for 13 h in a Beckman SW28 rotor at 4 °C. Gradients were fractionated using a Teledyne Isco gradient collector with pump speed set at 2 ml/min and chart speed at 30 cm/h. The fractions were collected in 13 ml round-bottom polypropylene tubes and transferred to 50 ml Nalgene Oakridge centrifuge tubes (3118-0050). Two volumes of guanidine-HCl 8 M and 3 volumes of 100% ethanol were added to samples for an overnight crude RNA precipitation at −80 °C. The samples were centrifuged at 20,000 × $g$ for 20 min at 4 °C. The pellets were washed with 3 ml of 85% ethanol and spun again for 20 min at 20,000 × $g$ at 4 °C. The pellets were dissolved in 3 ml of DEPC-treated water by vigorously vortexing 4 × 20 s. The resuspended RNAs were transferred to 13 ml polypropylene round-bottom tubes and precipitated with 300 μl of 3 M sodium acetate pH 5.3 and 2.5 volumes of 100% ethanol overnight at −80 °C and then centrifuged at 20,000 × $g$ for 20 min at 4 °C. The pellets were washed with 3 ml of 85% ethanol and spun again for 20 min at 20,000 × $g$ at 4 °C. The pellets were then dissolved in 1 ml of DEPC-treated water by vigorously vortexing 4 × 20 s. Proteins were removed by successive phenol:chloroform and chloroform extractions. 900 μl of aqueous phase was transferred to a 2 ml microtube and RNAs were precipitated with 1 ml of 3 M LiCl overnight at −80 °C. The RNAs were precipitated in a microcentrifuge at 13,000 × $g$ for 20 min at 4 °C and pellets were washed with 85% ethanol and spun again for 20 min at 13,000 × $g$ at 4 °C. The pellets were resuspended in 350 μl of DEPC-treated water. To remove LiCl, samples were precipitated again with 35 μl 3 M sodium acetate pH 5.3 and 1,155 ml of 100% ethanol overnight at −80 °C. The pellets were washed with 75% ethanol and dried in a speedvac at 40 °C for 20 min. The pelleted and dried pure RNAs were finally resuspended in 100 μl of DEPC-treated water and stored at −20 °C before deep sequencing analysis and RT-qPCR.

### RNA sequencing

Starting with 1 μg RNA from total, light polysome (40S + 60S + 80S) and heavy polysome (tetrasome and heavier) fractions, we enriched mRNA using the NEBNext Poly(A) mRNA Magnetic Isolation Module (E7490S). NGS libraries were then prepared with NEBNext using Ultra II Directional RNA Library Prep Kit for library construction in a poly(A) mRNA enrichment workflow (E7760S) according to the manufacturer's protocol. Individual libraries were purified with Ampure XP beads, analysed on an Agilent Bioanalyzer 2100 (HS DNA) for size and quality then quantified using Qubit 2.0. A first round (wild-type and *uL30bb* for light and heavy polysome fractions, untreated and staurosporine treated samples) of sequencing was performed on Illumina's NextSeq 500 sequencer. Two pools (one for each N) of 8 libraries/pool were sequenced in 2 separate runs using NextSeq 500/550 High output v2 kit (150 cycles) FC-404-2002 achieving a mean depth of 70 million reads per sample. A subsequent round (Wild-type and *ul30bb* total extracts; *ul30aa* total light, and heavy polysome fractions) of sequencing was performed with a pool of 20 samples for a mean depth of 30 million reads per sample. Two biological replicates of WT, R28P-30A and R28P-30B were run similarly as a pool of 18 samples.

### RT-qPCR

Contaminant genomic DNA from 5-25 μg of RNA, extracted either from total extracts or from sucrose gradient fractions, was removed by treatment with 33 units of Qiagen RNAse-free DNase (79254) on Qiagen RNeasy Minikit (74106) spin columns for 25 min at 37 °C. DNase was inactivated and washed away by the provided buffers and RNAs were eluted twice with 70 °C DEPC-treated water. A total of 50 ng DNase-treated RNA was used as template for reverse transcription using either RT Transcriptor from Roche and random hexamers or Moloney Murine Leukemia Virus-RT (MMULV-RT) locally produced by the protein purification service of Université de Sherbrooke. The qPCR reactions were carried out in 384 well plates in a Bio-rad CFX384™ Real Time System thermocycler in 10 μl volumes using the Bio-Rad iTaq Universal SYBR Green Supermix, 1 ng of cDNA and 200 nM of primers described in Supplementary Data 2.

### Protein preparation for mass spectrometry

Protein quantification by mass spectrometry was carried as previously described by[32] with the following modifications. Cells were grown and harvested as described in the polysome preparation section. For Figs. 1c, 2f and Supplementary Figs. 2d, f, 7b, 14b–d, the proteins were extracted in protein extraction reagent type 4 in 50 mM tris pH 8.0. For Fig. 2g and Supplementary Fig. 2c cells were grown and harvested and the sucrose gradients were performed as described in the polysome preparation section. To avoid interference, no heparin was added in extraction buffer nor in sucrose gradients. Spectra (3500 Da, Spectrum laboratories) tubing was used to dialyze the protein extracts against TSM buffer (10 mM Tris, 3 mM succinic acid, 10 mM MgCl$_2$, pH 8) over 70 h at 4 °C with continuous agitation and 6–8 changes of 5 litres of buffer. The dialyzed samples were frozen at −80 °C in 50 ml Nalgene Oakridge centrifuge tubes (3118-0050), lyophilized at low atmospheric temperature (150–250 mtorr) and resuspended in phosphate buffer (140 mM NaCl, 2.5 mM KCl, 4 mM Na$_2$HPO$_4$, 1.5 mM KH$_2$PO$_4$, pH 7.3). The samples were mixed by slow agitation at 4 °C with 0.4 volume of 1 M MgCl$_2$ and 2 volumes of glacial acetic acid, then centrifuged at 20,000 × $g$ for 10 min at 4 °C. A second dialysis was performed against 0.5% acetic acid over 70 h at 4 °C with continuous agitation and 6–8 changes of 5 l of buffer and lyophilized. Paralog-specific protein quantification was performed on LC-MS/MS TripleTOF 5600 mass spectrometer (ABSciex; Foster City, CA) equipped with a DuoSpray source at PhenoSwitch Bioscience (Sherbrooke, Canada). Reagent 4 (6 M urea, 2 M thiourea, 4% CHAPS in 50 mM Tris pH 8) was used to suspend lyophilized proteins and Pierce 660 protein assay used to

estimate the protein quantity. DTT was used to alkylate and reduce 40 μg of protein followed by overnight precipitation with 1 volume of ice-cold methanol and 8 volumes of ice-cold acetone at −80 °C. After centrifugation, pelleted proteins were washed 3 times with ice-cold methanol and briefly air dried. A first digestion was performed using 1:30 w/w ratio, of lysine C to proteins, followed by digestion using the same ratio of trypsin in 0.75 M urea and 50 mM Tris pH 8 for 4 h at 37 °C. Formic acid 2% v/v was used to stop digestion followed by peptide purification by solid-phase extraction on a polymeric reverse-phase cartridge (Phenomenex, Torrance, CA). LC-MS/MS. Digested and purified proteins were ionized by an ESI (Electron spray ionization) source in a 25 μm ESI probe (Eksigent) using a microLC200 system (Eksigent) equipped with a 150 mm × 300 μm HALO C18 2.7 μm column (Eksigent). Samples were injected by overfilling a 5 μl injection loop. A gradient of water containing 0.2% formic acid and 3% DMSO and ethanol containing 0.2% formic acid 3% DMSO were used to perform chromatography at 50 °C. Rolling collision energy and optimized SWATH windows in positive product ion mode with a mass range from 100 to 1800 m/z and in high sensitivity MS/MS mode were used for data acquisition. For Fig. 6b–d and Supplementary Fig. 16, proteins were extracted or resuspended in 8 M urea, 20 mM Tris pH 8. Samples containing 50 μg total proteins were treated with 5 mM DTT for 2 min at 95 °C, cooled down to RT for 30 min then treated with 7.5 mM chloroacetamide at RT for 20 min in the dark. Following dilution with 3 volumes of 50 mM $NH_4HCO_3$, samples were digested overnight at 37 °C with 1 μg of LysC (Promega)[40]. Samples were purified and desalted on 100 μl ZipTips (Pierce), eluted in 1% formic acid/50% acetonitrile, dried and resuspended in 30 μl of 1% formic acid. Following quantification at 205 nm, 1.5 μg of peptides were separated at the Plateforme de Protéomique (Université de Sherbrooke, Canada) on a Dionex Ultimate 3000 nanoHPLC system using an Acclaim PepMap100 C18 column (0.3 mm id × 5 mm, Dionex) followed by an EasySpray PepMap C18 nano column (75 μm × 50 cm, Dionex) with a 5–35% of 90% acetonitrile/0.1% formic acid over 240 min at a rate of 200 nl/min. Samples were ionized by an EasySpray source and transferred to an OrbiTrap Q Exactive mass spectrometer (Thermo Fisher Scientific). The spray voltage was set to 2.0 kV and the column temperature to 40 °C. Full scan MS survey spectra (m/z 350–1600) in profile mode were acquired in the Orbitrap with a resolution of 70,000 after accumulation of 1,000,000 ions. The ten most intense peptide ions from the preview scan in the Orbitrap were fragmented by collision-induced dissociation (normalized collision energy 25% and resolution 17,500) after the accumulation of 50,000 ions. Maximal filling times were 250 ms for the full scans and 60 ms for the MS/MS scans. Precursor ion charge state screening was enabled and all unassigned charge states as well as singly, 7 and 8 charged species were rejected. The dynamic exclusion list was restricted to a maximum of 500 entries with a maximum retention period of 40 s and a relative mass window of 10 ppm. The lock mass option was enabled for survey scans to improve mass accuracy. Data files were generated by the Xcalibur (v4.3.73.11) software.

### Total protein extraction and western blot analysis

Cells from exponentially growing 40 ml cultures were pelleted by centrifugation for 3 min at 3000 × g at 4 °C and washed with cold water. The pellets were resuspended in 500 μl of lysis buffer (20 mM Tris pH 8, 150 mM NaCl, 0.1% Triton X-100, 1 mM PMSF, 1 mM benzamidine, 1 μg/ml aprotinin, 1 μg/ml leupeptin, 1 μg/ml pepstatin A, 1 μg/ml antipain) and transferred to 2 ml tubes with 300 μl acid-washed glass beads. Cells were broken by 5 cycles of vortexing in a Bertin Precellys 24 homogenizer at 5000 rpm for 30 s followed by 30 s on ice. Lysates were cleared by centrifugation at 13,000 × g for 15 min at 4 °C and dosed by Bradford. Protein samples (20 μg) were separated on 15% SDS-PAGE and transferred to Protran nitrocellulose membranes (GE Healthcare). Membranes were blocked overnight at 4 °C with 5% milk in TBS-T

(20 mM Tris pH 7.6, 150 mM NaCl, 0.1% Tween-20), then incubated overnight at 4 °C with the following primary antibodies: rabbit anti-L7 (Bethyl Laboratories, cat#A3100-741A-M, 1:1000), rabbit anti-L5 (kindly provided by Woolford lab, 1:2000), mouse anti-L3 (kindly provided by the Warner lab, 1:5000) and mouse anti-Pgk1 (Invitrogen/Thermo Fisher, cat#459250, 1:10,000) diluted in blocking solution. After washing with 4 changes of TBS-T over 40 min, the membranes were incubated for 90 min at room temperature with secondary antibodies (donkey anti-rabbit HRP-IgG, GE Healthacare, cat#NA934V, 1:5000 or goat anti-mouse HRP-IgG, GE Healthcare, cat#NA931V, 1:2000) diluted in blocking solution and washed as before. Western Lightning Plus-ECL reagents (Perkin Elmer) were used for bioluminescence and a LAS4000 system (GE Healthcare) for data acquisition. The Quantity One software from Bio-Rad was used for the quantification. Unprocessed blots are available in the source data file.

### Quantification and statistical analysis

**RNA sequencing analysis.** Saturation and coverage tests were performed for all samples and coverage was found adequate in all cases. Sequencing reads were aligned on a genome reference sequence (R64/sacCer3) using the STAR aligner (default parameters) and all valid positions mapped were kept. Gene expression was quantified in TPM for the sequence generated from total RNA extracts or CPM for sequence used for the calculation of the translation index using featuresCount from the Rsubread package[41,42]. A summary of the sequencing data is presented in Supplementary Data 3.

**Proteomics analysis[43].** Data integration and analysis were initially performed by Peakview version 2.2. Peptide quantification carried out by Peakview was used for inter-sample normalization. Peak selection, protein quantification was performed using Skyline (MacCoss Lab). Relative paralog-specific quantification was performed without any normalization (Figs. 1c, 2f, g and Supplementary Fig. 2c, d, f), expression of uL30 from plasmids was normalized to large subunit RPs (Supplementary Fig. 14b), while protein quantification of genes over- and under-translated in *ul30bb* and RP quantification was done after normalization to Pgk1 protein (Supplementary Figs. 7b, 14d). For Figs. 4c and 6b–d and Supplementary Fig. 16, the raw files were analyzed using MaxQuant (v2.0.1.0) and Uniprot *S. cerevisiae* database (31/05/2021, 6079 entries). The settings used were: enzyme: LysC (K not before P); miscleavages: maximum of 4; fixed modifications: carbamidomethylation on C; variable modifications: carbamylation on K and N-termini, methionine oxidation and N-terminal acetylation; precursor ion mass tolerance of 7 ppm; fragment ion mass tolerance of 20 ppm; PSM FDR, protein FDR and site decoy fraction were set at 0.05. Peptides with the "Reverse" or "Potential contaminant" attributes were removed from the analysis. For Fig. 4c, the MaxQuant protein group table output of the wt and *ul30bb* samples were treated by LFQ analyst[44] with default settings and the Log2 FC was normalized to Pgk1. For Fig. 6b–d and Supplementary 16, the intensities of the N-terminal peptides specific for uL30A and uL30B were summed and classified as N-terminal acetylated or not. Percentage of acetylation or paralog ratios were calculated within samples, hence no normalization was applied. The detection of uL30 N-terminus was achieved by considering the 13 amino acid long peptide that results from the cleavage at the second lysine of ul30. The analysis took into consideration partial cleavage, and we verified that the efficiency of LysC digestion was similar across the samples at the proteome level.

**RT-qPCR analysis.** RT-qPCR were done in 2–3 biological replicates with 3 technical replicates. Relative quantification of mRNA was calculated according to the formula: $RQ = 2^{(CT_{Reference} − CT_{Test})_{Gene}}/2^{(CT_{Reference} − CT_{Test})_{Control}}$, where CTs are the cycle threshold values determined by the Bio-Rad CFX Manager software; Reference and Test represent strain, conditions or fractions compared; and Gene and

Control represent the gene of interest quantified and the normalization control gene. Inter-sample comparison was achieved by normalizing mRNA to Osh6 mRNA or Nme1 noncoding RNA as internal. The translation index was calculated by dividing the amount of mRNA in heavy polysome fractions (tetrasome and heavier fractions) by the amount of mRNA detected in light polysome fraction (40S, 60S, 80S).

**Gene ontology analysis.** The Gene Ontology (GO) Cellular Component enrichment analyses were carried out using the SGD server (https://www.yeastgenome.org/) with the default settings of GO Term Finder. Only the major categories of component gene ontology with significant $p$ value <0.001 using Bonferroni correction were considered.

**Kozak score analysis.** The Kozak score for 6717 transcripts were obtained by creating a matrix of the nucleotide frequencies (incidence of a nucleotide) for the six nucleotides upstream and three nucleotides downstream of the initiation codon AUG. A maximum score of 3.992 for the sequence ACAAAA-AUG-GCU was calculated.

**Statistical analysis**
Statistical tests and Ns are specified in figure legends and were performed using GraphPad Prism for Windows. Significance is given as $p$ value from a $t$ test for growth rates or mRNA and protein level comparisons, or from a Mann–Whitney test for genome-wide distribution comparisons as described in the figure legends.

**Reporting summary**
Further information on research design is available in the Nature Research Reporting Summary linked to this article.

## Data availability
The data that support this study are available from the corresponding author upon reasonable request. The RNA-seq data data generated in this study have been deposited in the NCBI Gene Expression Omnibus (GEO; https://www.ncbi.nlm.nih.gov/geo) under the accession number GSE133457 and GSE202803. The Proteomic data generated in this study have been deposited in the Peptide Atlas (http://www.peptideatlas.org/) under accession number PASS01404 and in the ProteomeXchange Consortium via the PRIDE partner repository with the dataset identifier PXD033843. All strains are available upon request. Source data are provided with this paper.

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

## Acknowledgements
We thank Jules Gagnon for help with the computational analysis of mRNA feature, Camille Martenon-Brodeur and and Kloé Neszvecsko for help with the preparation of yeast strains and the initial screen for paralog-specific effects, Julie Parenteau for advice and help with yeast genetics, Sonia Couture for help with RNA sequencing, Vincent Boivin and G.R. for the analysis of sequencing data. Sequencing was performed in the U de S Rnomics Platform. Proteomic analysis was performed at Phenoswitch and U de S Proteomic Platform. We thank John Woolford for providing anti-L5 antibodies and Jonathan Warner for providing anti-L3 antibodies. This work was supported by NSERC and a Research Chair in RNA Biology and Cancer Genomics (S.A.E.).

## Author contributions
M.M.G. and M.C. designed and performed experiments, analyzed data, produced figures, and participated in the writing of the paper. G.R. analyzed the RNA sequencing data. M.S. supervised the analysis of the RNA sequencing data and participated in the writing of the paper. S.A. conceived the project, designed experiments, analyzed data and wrote the paper.

## Competing interests
The authors declare no competing interests.
