## [Peer Review File · Nature Communications]

Duplicated ribosomal protein paralogs promote alternative translation and drug resistanceReviewers' Comments:

Reviewer #1:

Remarks to the Author:

Whether duplicated ribosomal protein (RP) genes may diversify ribosome composition and function is a substantial question in the field of translational gene regulation. Here, Malik-Ghulam et al conduct an in-depth functional characterization of one duplicated RP gene pair in yeast. The authors selected uL30A/B to investigate among the many duplicated RP genes in yeast, based on experiments showing that replacing the chromosomal sequence of uL30A with uL30B impairs growth but improves resistance to several drugs, including staurosporine. They go on to show that the uL30bb strain has preferential translation of a gene set that is enriched for long ORFs and cell periphery/wall/membrane genes. To show that these effects are independent of the 60S biogenesis defects seen in uL30bb, the authors replicate these findings using a different set of yeast strains in which the chromosomal copies of uL30A/B are deleted and replaced by a plasmid expressing uL30A, B, or A-B chimeric sequences driven by a constant promoter. Lastly, they show that uL30 A and B are differentially acetylated in an amino acid sequence-dependent manner.

A key strength of this work is that Malik-Ghulam et al make efforts to distinguish whether the phenotypes of their mutant strains are due to differences between uL30A and B in terms of expression level, 60S biogenesis role, chromosomal loci, ORF sequence, or RP amino acid sequence. The authors can firmly conclude that RP paralogues can produce ribosomes with different functional roles in translation control. This reflects a breakthrough finding in the field that has wide reaching implications. The data are extremely well performed, rigorous, and comprehensive. Most of the weaknesses of this manuscript can be amended with textual edits. I therefore recommend publication of this interesting and important manuscript with minor edits.

Major comments

Fig 6: While the differential acetylation of uL30A and B is intriguing, the evidence here cannot distinguish whether staurosporine resistance and preferential long ORF translation depend on acetylation, or on the N-terminal amino acid differences between uL30A and B. One way to investigate whether acetylation is at least correlated with ribosome activity (independent of amino acid sequence) is to show whether the ratio of acetylated vs. non-acetylated uL30A is different between light and heavy polysomes, and whether that changes upon staurosporine treatment. This is just a suggestion as this reviewer is unsure whether such an experiment is feasible.

Fig 6E: It is mentioned that the 5' portion of the uL30A/B ORFs were swapped both on plasmids and on chromosomal copies, but the referenced Fig S8 only seems to have data from the R28P-driven plasmids and it is unclear in Fig 6E whether the swapped 5' domain is done on the plasmid or chromosomal copy. Since a major claim of this work is that staurosporine resistance and preferential long ORF translation solely rely on the N-terminal acetylation and/or amino acid sequence of uL30A and B, it would be more robust to show whether these phenotypes are also observed in strains where only the differing amino acids have been substituted at the uL30A and B chromosomal loci. This would eliminate any question that the observed phenotypes are artifacts of exogenous RP expression from plasmids, 60S biogenesis defects, or perturbation of other elements in the chromosomal loci.

Discussion: can the authors cite literature supporting the assumption that translation of a transcript with less optimal codons and a weaker Kozak sequence (i.e. uL30B) will be less affected by stress than a highly translatable transcript (i.e. uL30A)? This is not intuitive and may depend on the particular type of stress and mode of translational deficiency, i.e. impaired initiation vs. elongation (see Ferretti and Karbstein 10.1261/rna.069823.118, and Acevedo et al 10.1038/s41598-018-22330-9). Other explanations, such as a stress-responsive element in the uL30B gene, may be possible.

Do the authors have thoughts on how acetylation or structural differences between uL30A/B might impact translation of long ORFs, or ORFs encoding cell wall/membrane/periphery proteins? The N-

terminus of uL30 appears to be solvent-accessible and distant from both the mRNA groove and the peptide exit tunnel. Since cell wall/membrane/periphery proteins are often secreted or integral membrane proteins synthesized at the endoplasmic reticulum (ER), the authors may wish to analyze whether other ER-translated genes are also overrepresented among genes that are up-translated in uL30bb or R28P-30B. Given the position of uL30 on the LSU, it could plausibly interact with ER-bound factors while the ribosome is bound to a translocon, and secondary structure or acetylation may modulate this.

Minor comments

- Fig 1D-F: where the legend uses solid and dashed lines for "major isoform" and "minor isoform" respectively, it would be more clear to label these as "expressing major isoform" and "expressing minor isoform". Currently, it is not clear from the figure alone whether the line style denotes strains that are still expressing an isoform, or those where that isoform has been deleted/edited.
- Pg. 6: "Extended Data Fig. 2a" referring to the wrong figure?
- Fig 4E: why is this experiment done with uL30a-del while most of the others are done with uL30bb?
- Fig 5D: Can authors address why R28P-30A is more resistant to staurosporine than WT, while uL30aa is less resistant (Fig 2B)?
- Fig 5K: Is the y-axis a log scale? Based on the definition of translation index in Fig 3A, it should not have negative values.
- Fig 6A: Main text says that uL30B has an N-terminal alanine and uL30A has a serine, but Fig 6A shows the opposite
- Fig 6E-F: Displaying the growth rates of WT, R28P-uL30A, R28P-uL30B, uL30B-5'A, uL30A-5'B, uL30B-Alike, and uL30A-Blike (which is not characterized in the manuscript) together in a single graph (+/- staurosporine) would be a more transparent and comprehensive way to demonstrate whether the 5' ends of uL30A/B or the 5 differing amino acids are necessary and/or sufficient for staurosporine resistance. As is, showing parts of these results across several different figures makes a direct comparison less straightforward.
- Figure S1B: missing decimal point in "587"
- Fig S2C: if uL30A is preferentially translated over uL30B under normal conditions, why does uL30bb express more total uL30 protein than WT? This does not seem to be explained by transcriptional upregulation - comparing the amounts of uL30 mRNA and protein in the uL30bb strain (Fig S2C) the mRNA:protein ratio seems similar between uL30aa and uL30bb.
- Fig S7E: Is it expected that the translation index for long mRNAs would be so different between uL30bb and R28P-30B, relative to WT? Is this because the proportion of polysomes to monosomes is so perturbed in uL30bb?

Reviewer #2:

Remarks to the Author:

NCOMMS-21-44443-T

In this manuscript Malik-Ghulam et al use a combination of quantitative growth assays, mass spectroscopy and genetic assays to decipher the role of the two Rpl30/uL7 isoforms in staurosporine resistance. The authors propose a model where the two isoforms are differentially acetylated and that the acetylation of Rpl30 regulates translation of mRNAs with long ORFs, which includes cell wall proteins, therefore mediating resistance to staurosporine and other drugs. The manuscript describes a set of interesting data. However, they do not fully support the conclusions put forward by the authors and are thus preliminary.

Major concerns:

1. I believe that the authors have not rigorously ruled out a model that the manipulations they do change ribosome concentrations and that these are what underlie the effects. Also, more globally the

changes in ribosome composition after staurosporine addition take 4 doublings to establish, not exactly a rapid stress response. This needs to be minimally acknowledged, as this is not a stress response, but perhaps a long-term adaptation. I do think the model that authors propose that translation status affects N-terminal acetylation is super interesting, as is the suggestion in the data (I also do not think this is rigorously shown), that 30A promotes ribosome assembly and thus normally A ribosomes are made even over the raw amounts of 30A and 30B.

2. The data are confusingly presented, as the data in the first half of the manuscript are in a system that the authors later (Figure 5) show produces effects that have nothing to do with translation but likely arise from effects on ribosome concentration and/or the depletion of other ribosomal proteins (shown in Extended data 2g). I would suggest these data be removed or presented in the supplement.

3. The data are entirely consistent with cells demonstrating reduced translation due to different ribosome concentrations being resistant to staurosporine. This would actually make sense given that staurosporine affects PKC and PKC affects translation initiation. It also explains the polysome profiles and growth defects/resistances. Moreover, it would also be consistent with the differences in acetylation. To better understand if this underlies the observations the authors should:

a. Measure ribosome abundance in all strains. A Northern will do. The mRNA levels are poorly correlated, as can be seen in S7 and S8. Panels S7d and S8e suggest that 30A-5'B has more ribosomes. This might account for all observations. Some data are shown in Figure 5f, but only ratios are shown. Brar and colleagues have shown that Rpl reductions lead to loss of Rps. This likely disguises effects here, and is essentially reproduced in Figure S2g. What the authors must show are 18S in all strains and 25S in all strains.

b. Show raw data for doubling times, not just normalized data. Doubling times in the range of 90-300 are very reliable, but higher numbers become less reliable. And most importantly, epistatic effects (as observed here) must be carefully controlled for, as it is possible that slower growth rates just cannot be observed (producing artifactual "epistasis"). Ie if you are doubling every 300 minutes it is difficult to slow down 2 times more....

c. Test if slowing translation by other means (partially phosphorylating eIF2, etc.) also produces staurosporine resistance.

d. Show polysome profiles with the relevant strains of Figure 5 and 6.

e. Ideally repeat the experiment from Figure 2e-g for the better controlled strains in Figures 5 and 6, but minimally show non-normalized data for panel g. It is important to know whether the A isoform moves out of the polysomes or the B isoform into the polysomes (or both).

f. The experiment in Figure 4e must be repeated with the "correct" strains of Figure 5 and 6. Also unnormalized data need to be shown. How does the Glt1 mRNA behave? And how does the Egd1 mRNA behave?

4. In Figure S2G the authors show that Rps25, uS4, and uL4 are dramatically reduced in the bb/aa strains. A few other RPs are also reduced more than the average, and the total ribosomes are lower in the BB strain. The authors need to reexamine this in the strains of Figure 5 and 6. They look at some proteins in Figure S7e, but not the ones they show here to be affected.

5. Related, the authors want to carefully consider and cite the paper by Johnson et al., (NAR 2020) about Rps25 loss.

6. I get the sense that the data in Figure 3 were also obtained with the "correct strains" of Figures 5 and 6. These should be shown in the main manuscript.

7. The data in Figure 5 are critical. The authors should show protein levels (not just mRNAs), as the Kozak sequence is not all that determines protein output; and absolute doubling times. As well as polysome profiles with staurosporine. The data as they are shown are entirely consistent with small differences in the 60S concentration, which could account for the data.

8. Figure 4f/5j: Given the correlation between TE and ORF length, I suspect that the TE in wt cells is also correlated with over and under translation. The authors should analyze this. Finally, the second part of the switch experiment, Short ORF into GLT1 locus is missing. This must be done with strains 9. I am confused about the data in Figure 6. According to the text panel b shows all proteins. But the label in the Figure indicates we are just looking at Rpl30A/B. The authors likely have the data for all proteins. They should be presented here.

10. Unnormalized data for panel 6c must be shown so the reader can evaluate which protein changes.
11. All panels need a wild type control, not just two mutants relative to each other. The data should in all cases be shown as unnormalized data in addition to whatever presentation the authors chose.
12. The data in Figure 6e and f do not support the contention of the author that it is simply the amino acid change in the N-terminus of Rpl30 that produces the effect. The data in Figure 5d are different from the data in 6e and f: Rpl30A and B in 5d show resistance. But the mutant in 6 shows sensitivity. Also, the panel should probably be labeled akin to Figure 5d, as that is the same type of data.
13. The authors suggest that the observed effects are mediated by N-terminal acetylation. If so, one would expect the Nat1 k/o to reproduce the effects of the knockout of the minor isoform.

Minor concerns:

1. If the authors want to retain the data in Figure 1, they should:
 - a. measure not just mRNA but also protein levels.
 - b. Show growth rates not just normalized but absolute. I am thinking about these very different if the doubling times with staurosporine are 150 min or 600 minutes. Same for doubling times of the strains without staurosporine.
 - c.
2. Figure 2e: Kap12 is not a 60S biogenesis protein. Dbp2 is a transcription factor, Rli1 is involved in 40S assembly, and Hep3 does not seem to be an actual gene in yeast.
3. The authors never probe translation, just polysome association. It is possible that polysome association increases because translation is slowed. They need to be careful in their wording. This also extends to the name "translation index". It should be called polysome binding index or something like that.
4. I would suggest they rename the uL30B-5'A isoform to: uL30B-A-Nt (A-N-TERMINUS), as they are indicating it is the protein, not the mRNA that matters.
5. It is unclear how uL30B alike differs from uL30B (or uL30B-5'A).
6. There are typos throughout the manuscript:
 - a. Abstract. Programed should be programmed
 - b. P.8 ribosome-associated mRNAs were sequenced. Not was.
 - c. Figure 7d axis title; protein is misspelled
7. Abstract: Here we demonstrate that gene duplication may serve. That's awkward. Either you demonstrate or it may be the case. In addition, the authors really only investigate one case uL30, and no generalizations should be drawn.
8. P3, bottom: Ferretti&Karbstein discuss RP concentration as a source of apparent functional specialization and provide evidence from a literature analysis that changes in their expression levels can account for apparent "resistance" effects.
9. P6. Secondary structure differences are not obvious from Figure S1.
10. P7. Staurosporine also reduces the polysomes in cells expressing uL30A and B. also, the paralog choice does affect the levels of other RPs. From both subunits.
11. Title and P.17: most N-acetylation is co-translational, not posttranslational. Indeed it is the point of the manuscript that changes are co-translational
12. Figure 4: 60S associated mRNAs are not a thing.
13. Figure 7 is extremely difficult to follow. The legend does not explain much.

REVIEWER COMMENTS

Reviewer #1 (Remarks to the Author):

Whether duplicated ribosomal protein (RP) genes may diversify ribosome composition and function is a substantial question in the field of translational gene regulation. Here, Malik-Ghulam et al conduct an in-depth functional characterization of one duplicated RP gene pair in yeast. The authors selected uL30A/B to investigate among the many duplicated RP genes in yeast, based on experiments showing that replacing the chromosomal sequence of uL30A with uL30B impairs growth but improves resistance to several drugs, including staurosporine. They go on to show that the uL30bb strain has preferential translation of a gene set that is enriched for long ORFs and cell periphery/wall/membrane genes. To show that these effects are independent of the 60S biogenesis defects seen in uL30bb, the authors replicate these findings using a different set of yeast strains in which the chromosomal copies of uL30A/B are deleted and replaced by a plasmid expressing uL30A, B, or A-B chimeric sequences driven by a constant promoter. Lastly, they show that uL30 A and B are differentially acetylated in an amino acid sequence-dependent manner.

A key strength of this work is that Malik-Ghulam et al make efforts to distinguish whether the phenotypes of their mutant strains are due to differences between uL30A and B in terms of expression level, 60S biogenesis role, chromosomal loci, ORF sequence, or RP amino acid sequence. The authors can firmly conclude that RP paralogues can produce ribosomes with different functional roles in translation control. This reflects a breakthrough finding in the field that has wide reaching implications. The data are extremely well performed, rigorous, and comprehensive. Most of the weaknesses of this manuscript can be amended with textual edits. I therefore recommend publication of this interesting and important manuscript with minor edits.

We appreciate the encouragement and positive feedback of this reviewer. Indeed, the reviewer lays out many of the reasons why this work merits timely publication in such a prestigious journal as Nature Communications.

Major comments

Fig 6: While the differential acetylation of uL30A and B is intriguing, the evidence here cannot distinguish whether staurosporine resistance and preferential long ORF translation depend on acetylation, or on the N-terminal amino acid differences between uL30A and B.

We agree with the reviewer that it is difficult to separate amino acid sequence and acetylation effects. We have tested and discussed these hypotheses accordingly in the new Results and Discussion sections.

One way to investigate whether acetylation is at least correlated with ribosome activity (independent of amino acid sequence) is to show whether the ratio of acetylated vs. non-acetylated uL30A is different between light and heavy polysomes, and whether that changes upon staurosporine treatment. This is just a suggestion as this reviewer is unsure whether such an experiment is feasible.

The reviewer had insight here and we followed their suggestion and examined the percent acetylation of uL30 paralog in both light and heavy polyribosome fractions. As indicated in Supplementary Fig. 15, the acetylated forms of uL30 were more abundant in the heavy polyribosome than in the light fraction. In the case of uL30B, 100% of the proteins detected in the heavy polyribosome fraction are acetylated, while only 65% of

the proteins in the light fractions were acetylated. Indeed, the non-acetylated form of uL30B is too scarce to detect in total extract and could only be detected in the light polyribosome fraction (Fig. 6c and Supplementary Fig. 15). In the case of uL30A, which is mostly non-acetylated, 8% of the proteins detected in the heavy polyribosome were acetylated and only 2% of those detected in the light fraction were acetylated. Furthermore, exposure to staurosporine did not change the percent acetylation of A and B forms in either total extract or the heavy polyribosome fraction (Supplementary Fig. 15). Instead, exposure to staurosporine increased the overall number of acetylated ribosomes by increasing the number of ribosomes incorporating the mostly acetylated B form of uL30 (Figs. 2g and 6c). This is consistent with our model suggesting that the change in ribosome acetylation, in response to staurosporine, is triggered by changes in the number of ribosomes incorporating the mostly acetylated uL30B (Fig. 7).

Fig 6E: It is mentioned that the 5' portion of the uL30A/B ORFs were swapped both on plasmids and on chromosomal copies, but the referenced Fig S8 only seems to have data from the R28P-driven plasmids and it is unclear in Fig 6E whether the swapped 5' domain is done on the plasmid or chromosomal copy.

The mutation in the chromosome had a similar effect to those performed in the plasmid when the appropriate comparison is made between isogenic strains. As requested, the data is now shown in Fig. 6E and Supplementary Fig. 16C. The Experiment in Fig 6E was performed using mutations in the chromosomal copy.

Since a major claim of this work is that staurosporine resistance and preferential long ORF translation solely rely on the N-terminal acetylation and/or amino acid sequence of uL30A and B, it would be more robust to show whether these phenotypes are also observed in strains where only the differing amino acids have been substituted at the uL30A and B chromosomal loci. This would eliminate any question that the observed phenotypes are artifacts of exogenous RP expression from plasmids, 60S biogenesis defects, or perturbation of other elements in the chromosomal loci.

We added the effect, of the chromosomal mutations of the N-terminal domain, on the ribosome profile and translation, as requested, in the new Supplementary Fig. 6f and h. The new data indicate, once again, that mutation of the N-terminal domain changes resistance to staurosporine and translation without affecting the overall ribosome profile.

Discussion: can the authors cite literature supporting the assumption that translation of a transcript with less optimal codons and a weaker Kozak sequence (i.e., uL30B) will be less affected by stress than a highly translatable transcript (i.e. uL30A)? This is not intuitive and may depend on the particular type of stress and mode of translational deficiency, i.e. impaired initiation vs. elongation (see Ferretti and Karbstein 10.1261/rna.069823.118, and Acevedo et al 10.1038/s41598-018-22330-9). Other explanations, such as a stress-responsive element in the uL30B gene, may be possible.

As suggested by the reviewer, the difference in translation of uL30 paralogs in response to staurosporine is likely driven by differences in translation efficiency. This is supported by previous reports showing that the more efficiently translated major paralogs of ribosomal proteins (e.g., uL30A) are more likely to be affected by stress than the less translated minor paralogs (see Malik et al., NAR 2020 Feb 28; 48(4): 1954–1968). As

requested, we have added a paragraph in the discussion describing the literature on effects of translation efficiency on gene response to stress.

Do the authors have thoughts on how acetylation or structural differences between uL30A/B might impact translation of long ORFs, or ORFs encoding cell wall/membrane/periphery proteins? The N-terminus of uL30 appears to be solvent-accessible and distant from both the mRNA groove and the peptide exit tunnel. Since cell wall/membrane/periphery proteins are often secreted or integral membrane proteins synthesized at the endoplasmic reticulum (ER), the authors may wish to analyze whether other ER-translated genes are also overrepresented among genes that are up-translated in uL30bb or R28P-30B. Given the position of uL30 on the LSU, it could plausibly interact with ER-bound factors while the ribosome is bound to a translocon, and secondary structure or acetylation may modulate this.

Taking this suggestion, we analyzed the effect of the paralog's mutation on ER-translated genes in Supplementary Fig. 9. Interestingly, as the reviewer predicted, we found that expression of the minor paralog of uL30B indeed favors the translation of ER-associated genes, further bolstering our cell periphery gene hypothesis. This finding is also consistent with the proximity between uL30, and ER-bound factors, as noted by the reviewer.

- Fig 1D-F: where the legend uses solid and dashed lines for "major isoform" and "minor isoform" respectively, it would be more clear to label these as "expressing major isoform" and "expressing minor isoform". Currently, it is not clear from the figure alone whether the line style denotes strains that are still expressing an isoform, or those where that isoform has been deleted/edited.

Our thanks to the reviewer for this comment. We have modified the label to Major Expressed and Minor Expressed to indicate the identity of the expressed isoform and further explained it clearly in the legend.

- Pg. 6: "Extended Data Fig. 2a" referring to the wrong figure?

We corrected the erroneous reference to Supplementary Fig. 2a to Supplementary Fig 3a

- Fig 4E: why is this experiment done with uL30a-del while most of the others are done with uL30bb?

We performed the chromosomal swap of the long and short ORFs in the uL30a Δ strain because it displays the same translation profile as uL30bb but is much easier to manipulate genetically. We have compared those two strains carefully and found no distinguishable differences.

- Fig 5D: Can authors address why R28P-30A is more resistant to staurosporine than WT, while uL30aa is less resistant (Fig 2B)?

We have now addressed this observation by adding a comparison between the acetylation levels of plasmid- and chromosomal-borne copies of uL30A and uL30B in Supplementary Fig. 15c. The difference between staurosporine resistances of uL30aa and R28P-30A is likely due to differences in the gene expression and the resulting change

in acetylation pattern. Expression from plasmids increases the amount of uL30A mRNA and changes the 5'UTR sequence which may alter the translation pattern of the plasmid borne proteins. Indeed, we found that the N-terminal acetylation of uL30A increases when it is expressed from a plasmid, making it more similar to uL30B acetylation levels. This observation supports our conclusion that the difference in the paralogs' resistances to staurosporine is affected by differences in the expression and acetylation pattern of the duplicated genes.

- Fig 5K: Is the y-axis a log scale? Based on the definition of translation index in Fig 3A, it should not have negative values.

Yes, the Y-Axis is in log scale. For clarity we now changed the title of the y-axis in Fig. 5K to "Translation Index Log₂FC Difference". The data display the log₂FC difference in the association of RNA with different sizes, with heavy and light polyribosomes in cells expressing different forms of uL30. Negative values therefore represent mRNA size categories that are more associated with heavy polyribosomes in cells expressing the A form and positive values represent mRNA size categories that are more associated with heavy ribosomes in cells expressing the B form.

- Fig 6A: Main text says that uL30B has an N-terminal alanine and uL30A has a serine, but Fig 6A shows the opposite

We corrected the typo in the text. The figure remains correct; uL30A contains alanine and B serine.

- Fig 6E-F: Displaying the growth rates of WT, R28P-uL30A, R28P-uL30B, uL30B-5'A, uL30A-5'B, uL30B-Alike, and uL30A-Blike (which is not characterized in the manuscript) together in a single graph (+/- staurosporine) would be a more transparent and comprehensive way to demonstrate whether the 5' ends of uL30A/B or the 5 differing amino acids are necessary and/or sufficient for staurosporine resistance. As is, showing parts of these results across several different figures makes a direct comparison less straightforward.

We have added a new Supplementary Table 4 that compares the doubling time of all uL30 strains used in this study, in the presence and the absence of Staurosporine, and illustrated it in Supplementary Fig. 17b.

- Figure S1B: missing decimal point in "587"

The error was corrected, with thanks to the reviewer.

- Fig S2C: if uL30A is preferentially translated over uL30B under normal conditions, why does uL30bb express more total uL30 protein than WT? This does not seem to be explained by transcriptional upregulation – comparing the amounts of uL30 mRNA and protein in the uL30bb strain (Fig S2C) the mannoproteins ratio seems similar between uL30aa and uL30bb.

The differences in the translation of uL30A and uL30B is only observed in wild type cells when the two gene copies compete. Indeed, it should be noted that uL30B is less translated only by RPG standard and only when outcompeted by uL30A, but otherwise

better or as well translated than most non-ribosomal genes including actin. In the absence of uL30A, the abundance and translation of uL30B mRNA increases leading to increased protein production to compensate for the loss of uL30A. This over-compensation for the loss of the major paralog is a general feature of most dRPGs and was previously demonstrated in the literature (Parenteau et al., Cell Rep. 2015, 13:2516 and Petibon et al., NAR 2016, 44:3878 and Ghulam et al., 2020, 48:1954).

- Fig S7E: Is it expected that the translation index for long mRNAs would be so different between uL30bb and R28P-30B, relative to WT? Is this because the proportion of polysomes to monosomes is so perturbed in uL30bb?

The difference between the translation index detected in cells expressing uL30 from plasmid-borne genes and those expressing it from chromosomal copies is indeed expected given the difference in the paralog expression, acetylation pattern and ribosome profile. However, this difference does not mean that the capacity of the A and B forms to support the translation of long ORFs is linked to the ribosome profile. The difference in translation index is largely reduced and practically disappeared for genes longer than 4000 nt when we compare the translation index of isogenic strains expressing the paralogs from either plasmids or chromosomal copies (Fig. 5k). This indicates that similar differences in paralog dependent translation as that observed with the chromosomal copies (ul30aa and ul30bb) could also be observed when the plasmid strains (R28P-30A and R28P-30B) are compared to each other despite their normal ribosome profile.

Reviewer #2 (Remarks to the Author):

NCOMMS-21-44443-T

In this manuscript Malik-Ghulam et al use a combination of quantitative growth assays, mass spectroscopy and genetic assays to decipher the role of the two Rpl30/uL7 isoforms in staurosporine resistance. The authors propose a model where the two isoforms are differentially acetylated and that the acetylation of Rpl30 regulates translation of mRNAs with long ORFs, which includes cell wall proteins, therefore mediating resistance to staurosporine and other drugs. The manuscript describes a set of interesting data. However, they do not fully support the conclusions put forward by the authors and are thus preliminary.

We thank this reviewer for the time and effort put into reviewing this manuscript and for finding the data interesting.

We should point out immediately that we did not set out to prove that differences in acetylation and amino acid sequence of the duplicated ribosomal protein genes are directly and solely responsible for modifying translation. Rather, we set out to understand the general contribution of dRPG to cell growth and understand the cellular function of paralog sequences and expression patterns. With this view in mind, we generated a systematic study that dissects different aspects of the paralog expression patterns and function including difference in the paralogs' support of ribosome biogenesis and their differential downstream impact on translation and cell growth. The exact mechanism by which acetylation and/or amino acid differences can affect translation is clearly beyond the scope of this in vivo study. However, to address this reviewer's concerns we have now added new data revealing that paralog-specific differences in translation and staurosporine resistance occur without differences in ribosome number (see answer 1 below). In addition, because of the reviewer's main concern we have now limited our Conclusion to the direct experimental evidence while mentioning alternative possibilities of the paralogs' effects on translation in the Discussion, with a view to accommodate the reviewer's comments without deviating from the original goal of this study, which was to characterize paralog-specific cell growth phenotypes.

1. I believe that the authors have not rigorously ruled out a model that the manipulations they do change ribosome concentrations and that these are what underlie the effects.

The effect of ribosome concentration is the predominant recurring theme of this reviewer. Please see the answer to question 3a below for the most accessible refutation of this possibility. To address the reviewer's concern regarding the ribosome concentration, we added new data comparing the subunit ratios and the rRNA amounts in all strains (Supplementary Fig.17a). We do not see reduction in the rRNA amount in any of the strains other than those expressing the minor chromosomal copy as the sole source of uL30, namely ul30a Δ and ul30bb. Whenever uL30 is expressed from the chromosomal uL30B gene we see clear defects in the biogenesis of the 60S subunits. Any small changes other than those observed in these two strains (ul30a Δ and ul30bb) are variable changes generated from the unavoidable, culture-to-culture and extraction day, variations

in RNA quality. Please note that in addition, we only ever compared isogenic strains with one another and did not compare independently generated strains with different genotypes (e.g., plasmid vs chromosomal copies). This avoids any strain background difference that could change the ribosome biogenesis and function baseline.

Also, more globally the changes in ribosome composition after staurosporine addition take 4 doublings to establish, not exactly a rapid stress response. This needs to be minimally acknowledged, as this is not a stress response, but perhaps a long-term adaptation.

All RNA and translation analyses were performed after 2 generations not 4. That said, it is important to note that the term “response to stress” was meant to convey the adaptive changes occurring after exposure to staurosporine and not to imply short-term stress response. To address the reviewer’s concern, we changed the passage to “adaptation to drug treatment”.

I do think the model that authors propose that translation status affects N-terminal acetylation is super interesting, as is the suggestion in the data (I also do not think this is rigorously shown), that 30A promotes ribosome assembly and thus normally A ribosomes are made even over the raw amounts of 30A and 30B.

We thank this reviewer for finding the data interesting and appreciate the encouragement although the requirement to rigorously prove our speculations as well as our inferences is beyond the scope of our study at this point. The main inference of our manuscript is that dRPGs differentially affect ribosome biogenesis and function independent of gene dose and that this difference in uL30 paralog function depends on the N-terminal sequences. The data also support the conclusion that the acetylated amino acid residues are required for staurosporine resistance. Our data only show that the expression of the two chromosomal copies of uL30B cannot compensate for uL30A dependent defect ribosome biogenesis. Showing that uL30A makes more ribosome than uL30B in wild type cells is difficult since all uL30B that is not incorporated into ribosome is degraded. However, to comply with the reviewer’s high standard for accuracy, we did change the word “promote” to “required”. We also clarified the statements across the text.

2. The data are confusingly presented, as the data in the first half of the manuscript are in a system that the authors later (Figure 5) show produces effects that have nothing to do with translation but likely arise from effects on ribosome concentration and/or the depletion of other ribosomal proteins (shown in Supplementary 2g). I would suggest these data be removed or presented in the supplement.

We assume that the reviewer is referring to the technique of gene homogenization that we use in the first half of the manuscript to explore the requirement, for expression of each locus, for cell and ribosome function. We believe that this technique is very important for the following reasons: 1) It creates genetically clean yeast strains that differ only in gene version and not gene dose and as such it gives us a holistic view of duplicated gene function, 2) Full gene homogenization allows us to identify both regulatory and coding sequence causing difference in paralog functions, 3) It allows us to mimic the original event of gene duplication and the natural situation where cells shut down the expression of one locus in favor of the other under different conditions, 4) it helps us avoid artifacts

generated by creating chimeric genes that incorporate elements from different genes that are normally not found together in nature. Without this elegant genetic approach, we could not have discovered the requirement of the A form for ribosome biogenesis, which is lost when the genes are expressed outside their chromosomal copies.

We believe also that our data are logically presented because in the absence of prior knowledge, of which part of the gene needs to be mutated, we had to first resort to whole gene homogenisation before we could dissect the etiology of the paralog-specific effects, in the second part of the manuscript, using plasmid-borne proteins to bolster our main conclusion that the difference in uL30 paralogs stems from differences in the sequence of the N-terminus rather than on protein concentration. The effect on translation we discover, while an interesting and exciting biproduct of the elegant systematic approach presented in the first half of the manuscript, should not occlude the importance of the main conclusion that differences in paralog functions are not simply caused by differences in quantities of mRNA and protein, produced by each of the two gene copies. Finally, we believe that arriving at the same conclusion using a completely different expression system and genetic background (e.g., gene homogenization, plasmid borne genes, mutation in plasmid borne genes and mutation in the chromosomal loci) significantly increases the robustness of our study and validates the conclusion and addresses the needs of readers interested in different aspects of yeast genetics, ribosome biogenesis, evolution of duplicated genes and translation. Because of these considerations we prefer to keep the data in the main manuscript.

3. The data are entirely consistent with cells demonstrating reduced translation due to different ribosome concentrations being resistant to staurosporine. This would actually make sense given that staurosporine affects PKC and PKC affects translation initiation. It also explains the polysome profiles and growth defects/resistances. Moreover, it would also be consistent with the differences in acetylation.

Concentration cannot explain all this as the data in Fig. 5, Fig. 6, and Supplementary Fig. 16 and 17 shows no correlation between ribosome number and staurosporine resistance. Indeed, plasmid-borne paralogs show difference in translation and staurosporine resistance without any statistically significant variation in growth rate or ribosome number. There is no doubt from these figures that staurosporine resistance could be achieved without significant reduction in ribosome amount, growth rate or 60S subunit. That being said, we do not necessarily exclude the possibility that changes in ribosome concentration could, under certain situation (e.g., uL30bb), also contribute to resistance to staurosporine as the reviewer suggested but this does not refute the demonstration of achieving staurosporine resistance without changes in ribosome concentration. To address the reviewer's comment, we now added a discussion of these different possibilities.

To better understand if this underlies the observations the authors should:

a. Measure ribosome abundance in all strains. A Northern will do. The mRNA levels are poorly correlated, as can be seen in S7 and S8. Panels S7d and S8e suggest that 30A-5'B has more ribosomes. This might account for all observations. Some data are shown in Fig.5f, but only ratios are

shown. Brar and colleagues have shown that *Rpl* reductions lead to loss of *Rps*. This likely disguises effects here, and is essentially reproduced in Fig. S2g. What the authors must show are 18S in all strains and 25S in all strains.

The differences in mRNA amounts, observed in Supplementary Fig. 13a and 16d, are not statistically significant and they reflect small variations in RNA extraction that do not translate into differences in protein amounts. Furthermore, the direction of these small variations further supports our model. For example, in Fig. 12a, the only difference between the isogenic strain pairs being compared is found between *ul30bΔ* and *ul30aΔ*. In this case, the strain lacking the major isoform (*ul30aΔ*) expresses slightly more RNA than the strain lacking the minor isoform (*ul30bΔ*), which strongly supports our conclusion that the phenotypic defect observed with *ul30aΔ* is not due to mRNA dose. Similarly, small variations in ribosomal proteins (new Supplementary Fig. 13d) reflect the random noise created by differences between biological replicates. In Supplementary Fig. 16f, we do not observe any differences in total ribosome number, but rather a slight increase in heavy polyribosomes, in strains expressing uL30 protein containing the minor paralog N-terminal domain and the major paralog C-terminus (R28P-30A-Bnt), which is consistent with our conclusion that the minor paralog N-terminal domain increases association with long mRNA. In all cases, the small variations, noticed by the reviewer, being statistically insignificant, further supports our conclusion that staurosporine resistance is not linked to reduced uL30 or ribosome amount. However, to fully comply with the reviewer's request, we have quantified the 18S and 25S rRNA in all strains using capillary electrophoresis and included the data in Supplementary Fig. 16e. In addition, we now include the raw data for the rRNA ratio comparison in Supplementary figures and tables. As the new comparisons (Supplementary Fig. 17) further confirm, we do not see any significant difference in the 18S or 25S between isogenic strains, except those already noted in Fig. 2 as defective in 60S biogenesis. It is now perfectly clear then that we see staurosporine resistance without any significant differences in ribosome number. As indicated in the summary figure at the top of this answer (Detailed in Supplementary Fig. 16), R28P-30B strains do not display any decrease in the 18S and 25S rRNA or the polyribosome profile and yet they show significantly more resistance to staurosporine. This data directly supports the main conclusion of the paper and clearly indicates that there are differences in translation and drug resistance without any significant defects in ribosome number or biogenesis.

Staurosporine resistance is not linked to reduced ribosome number. The first panel compares the amount of the 18S and 25S. The polyribosome profile, staurosporine resistance and effect on translation are showing the second, third and fourth panel respectively. More comparisons are indicated in Supplementary Fig. 16.

b. Show raw data for doubling times, not just normalized data. Doubling times in the range of 90-300 are

very reliable, but higher numbers become less reliable. And most importantly, epistatic effects (as observed here) must be carefully controlled for, as it is possible that slower growth rates just cannot be observed (producing artifactual “epistasis”). If you are doubling every 300 minutes it is difficult to slow down 2 times more....

The doubling times of all strains are now shown in Table 4 and illustrated in Supplementary Fig. 17b. As indicated on the right all doubling times vary between 80-240 minutes.

c. Test if slowing translation by other means (partially phosphorylating eIF2, etc.) also produces staurosporine resistance.

Generic decrease in translation resulting from slow growth rate is not sufficient for staurosporine resistance. As indicated in new Supplementary Fig 4, deletion of *ARP5* (actin related protein) and *PDX3* (pyridoxine phosphate oxidase) genes slows growth with no effect on staurosporine resistance. That is said, we have evidence that other ribosomal protein mutations, that slow growth and affect ribosome function, increase staurosporine resistance, as in the case of eL27 paralogs (Fig. 1f). This is consistent with our model suggesting that the expression of uL30B leads to staurosporine resistance by attenuating or slowing translation. Reducing the activity of the same number of ribosomes or reducing the overall number of ribosomes may lead to similar effects. We are not denying that the minor isoform may slow translation, but we argue it could do that without reducing the overall number of ribosomes produced.

d. Show polysome profiles with the relevant strains of Figure 5 and 6.

The polysome profiles were added as requested in Fig. 5 and 6.

e. Ideally repeat the experiment from Figure 2e-g for the better controlled strains in Figures 5 and 6, but minimally show non-normalized data for panel g. It is important to know whether the A isoform moves out of the polysomes or the B isoform into the polysomes (or both).

The experiments shown in Fig. 2e-g were performed in wild-type cells and as such are the best cells for this type of experiment. As requested, we added the non-normalized sequencing data for Fig. 2e in Supplementary Fig. 5 showing that it is the number of reads of the uL30B mRNA that increase in the heavy polyribosome fraction leading to a change in its translation index (heavy over light mRNA). For panel f-g only relative data could be shown since they are produced MRM data in which only uL30A and B were quantified and compared to each other. As such the data in panels f and g are the only valid direct comparison possible for this type of experiment.

f. The experiment in Figure 4e must be repeated with the “correct” strains of Figure 5 and 6. Also unnormalized data need to be shown. How does the Glt1 mRNA behave? And how does the Egd1 mRNA behave?

The experiment in Fig. 4e was performed in the correct strain since we are directly testing the conclusion reached in Fig. 2a using both wild-type and the deletion strain showing the strongest difference in translation of long and short genes. Carrying this mutation in the wild-type strain is the best way to show the staurosporine effect depends on ORF length. Repeating this experiment in strains artificially expressing the paralog from plasmids will not bring more information, since we already demonstrated that long and short ORFs are differentially expressed in those strains also (Fig 5k and Supplementary Fig. 16g). The unnormalized data of Figs. 4, 5k and 6c are now shown in Supplementary Figs. 12, 14 and 16 respectively.

4. In Figure S2G the authors show that Rps25, uS4, and uL4 are dramatically reduced in the bb/aa strains. A few other RPs are also reduced more than the average, and the total ribosomes are lower in the BB strain. The authors need to reexamine this in the strains of Figure 5 and 6. They look at some proteins in Figure S7e, but not the ones they show here to be affected.

We re-examined ribosomal protein quantities in all strains, as requested, and found no significant differences except, as would be expected, in strains with less 60S ribosomal subunit. Indeed, in previous Supplementary Fig. 2G we had not seen the difference in total protein amount but rather differences in paralog ratio. We have replaced this figure with new Supplementary Fig. 2f where RPs were quantified using peptides common to the paralogs. The data now show clearly show that none of the ribosomal proteins are significantly less abundant in cells expressing the minor paralog.

5. Related, the authors want to carefully consider and cite the paper by Johnson et al., (NAR 2020) about Rps25 loss.

The study of Rps25 deletion reported by Johnson et al., was performed in mammalian cells and the type of rewiring, adaptation and effects described in this paper are less likely in yeast. We took this work into account and measured the effect of our deletion and mutation in freshly transformed cells and limited the number of generations to 20 to avoid any long-term genetic adaptation. Furthermore, we tested all strains to ensure that we had successfully rescued and reproduced the paralog-specific phenotype by expressing the deleted genes from plasmids. The effect of deleting uL30 genes could be fully restored by expressing the deleted copy from a plasmid. Furthermore, we directly compared the effect of A and B expressed from plasmids in otherwise identical deletion strains (please see Supplementary Fig. 16). We believe our new work provides clear justification for the importance of repeating the experiment in different strains and not depending on any single mutation or deletion, to avoid the problem reported in the Johnson et al., work. In all cases, we thank the reviewer for bringing this paper up since it helps us demonstrate the quality and importance of our current work. We have cited this paper in the discussion to emphasize the importance of the systematic approach used in our study, which starts with classical genetics and gene homogenization and finishes with plasmid and targeted mutation-based analysis.

6. I get the sense that the data in Figure 3 were also obtained with the “correct strains” of Figures 5 and 6. These should be shown in the main manuscript.

Yes, the essential most pertinent data in Fig. 3 were also produced using the same chromosomal strains and included in Fig. 5 and Supplementary Fig. 16.

7. The data in Figure 5 are critical. The authors should show protein levels (not just mRNAs), as the Kozak sequence is not all that determines protein output, and absolute doubling times. As well as polysome profiles with staurosporine. The data as they are shown are entirely consistent with small differences in the 60S concentration, which could account for the data.

The data in fig 5e are not consistent with differences in 60S. We do not see any difference in the 60S subunit with the plasmid containing strains, as evident from the polyribosome profile. We have also further checked this with a more complete comparison of doubling time, protein amount and ribosome subunit quantification, using rRNA as requested, in Supplementary Figs. 13e and 17.

8. Figure 4f/5j: Given the correlation between TE and ORF length, I suspect that the TE in wt cells is also correlated with over and under translation. The authors should analyze this.

As requested also, we compared the translation efficiency of the different genes that are over- and under-translated in cells expressing the B form and found no correlation between the change in translation index of the long ORFs and their translation efficiency (Supplementary Fig. 11). The new figure indicates that long ORFs that are better translated in ul30bb cells are not necessarily under translated in wild-type cells.

Finally, the second part of the switch experiment, Short ORF into GLT1 locus is missing. This must be done with strains

Swapping short ORFs with longer versions is difficult as the integration efficiency is significantly reduced with insert sizes above 3 Kb. Despite several attempts, we could not get chromosomal insertion of the 6.42Kb GLT1 ORF. In addition, we tried with other genes with similar length without success. In any, case, the results of this experiment alone would not change the overall conclusion. The importance of the ORF length is already demonstrated by the translation profiles in Fig.4a and b and the swap of the long ORF with short one (Fig. 4e). Doing the reciprocal experiment (i.e., swapping the short ORF with long one will not change the fact that ORF length is important, as it can only provide additional evidence that the ORF length is the only factor, which we do not necessarily claim, or feel is important for the conclusion of this paper. We made sure to restrict our interpretation to the effect of decreasing, and not increasing, the ORF length on the translation of the mutated gene in response to changing the expressed uL30 paralog and exposure to staurosporine.

9. I am confused about the data in Figure 6. According to the text panel b shows all proteins. But the label in the Fig. indicates we are just looking at Rpl30A/B. The authors likely have the data for all proteins. They should be presented here.

We did not check the acetylation of all proteins, which is clearly beyond the scope of this paper. Neither do we have a reason to look for other protein acetylation events, as our study focuses only on differences between uL30 paralogs. Fig. 6 reports only uL30 acetylation, and we corrected the text accordingly. Panel b of the Figure compares the acetylation of the A and B form of uL30 and in the text “all proteins” refers to all uL30 proteins produced in the cells. We clarified this point in the text.

10. Unnormalized data for panel 6c must be shown so the reader can evaluate which protein changes.

The percentages of N-terminal acetylation of each paralog in each fraction are now included in Supplementary Fig. 15. As previously reported, the data clearly indicate the increased representation of N-terminal acetylated uL30 in heavy polysomes. The actual raw peptide intensities cannot be appropriately compared between different fractions without normalization.

11. All panels need a wild type of control, not just two mutants relative to each other. The data should in all cases be shown as unnormalized data in addition to whatever presentation the authors chose.

All data are either performed in wild-type cells or presented relative to wild-type control. In Fig. 6c the experiments were performed in wild-type cells. In Fig. 6d uL30A and uL30B data are produced from wild-type cells and compared with the N-terminal swap.

12. The data in Figure 6e and f do not support the contention of the author that it is simply the amino acid change in the N-terminus of Rpl30 that produces the effect. The data in Figure 5d are different from the data in 6e and f: Rpl30A and B in 5d show resistance. But the mutant in 6 shows sensitivity. Also, the panel should probably be labeled akin to Figure 5d, as that is the same type of data.

We have clarified the data in the new version of the figure. The data in new Figure 6e now clearly indicate that changing the N-terminus sequence of the chromosomal copy of uL30B with that of uL30A decreases resistance to staurosporine, when compared to wild type strain. These data clearly indicate that the N-terminal domain of uL30B is required for maintaining the normal tolerance to staurosporine, even when the other copy uL30A is not altered. In Fig. 5d, we test staurosporine resistance of strains containing plasmid expressing either uL30A or uL30B as the sole source of uL30 in the cells (these strains carrying double deletion of the natural uL30A and uL30B genes). In this case, once again we show that cells expressing uL30B are more resistant than those expressing uL30A, which is the main conclusion of the paper and is perfectly consistent with the decreased resistance observed with the mutation of uL30B N-terminus domain shown in Fig. 6. The increased resistance of cells expressing the major paralog from plasmid (R28P-30A) relative to wild type (Fig. 5d) is probably due to the change in the expression pattern of uL30A when expressed from plasmid, which makes it more like uL30B than when expressed from the chromosomal copy (Supplementary Figs. 6 and 15c). In fact, this observation further supports our conclusion that the acetylation level may contribute to staurosporine resistance since the increased acetylation of uL30A produced from plasmid associates with increased resistance albeit not to the same level as cells expressing uL30B. As requested, the axis title of Fig. 6e has been renamed akin to Fig. 5d.

13. The authors suggest that the observed effects are mediated by N-terminal acetylation. If so, one would expect the Nat1 k/o to reproduce the effects of the knockout of the minor isoform.

Nat1 knockout will have a pleiotropic effect caused by the altered acetylation of different proteins and cause defects in respiratory growth rate, reduce survival in stationary phase, and decrease cell fitness. As such, it would be impossible to ascertain the origin of any phenotype obtained with this knockdown.

Minor concerns:

*1. If the authors want to retain the data in Figure 1, they should:
a. measure not just mRNA but also protein levels.*

We previously demonstrated that translation and protein amounts of most dRPGS (including those examined in Fig. 1) are directly correlated with the level of mRNA produced by the genes (Malik et al., NAR 2020 PMID: 31863578) and we have referred to this data in the revised manuscript.

b. Show growth rates not just normalized but absolute. I am thinking about these very different if the doubling times with staurosporine are 150 min or 600 minutes. Same for doubling times of the strains without staurosporine.c.

The growth rates are now included in Supplementary Table 4 and Supplementary Fig. 17. Untreated doubling times vary from 1.3h to 2.8h, measured doubling times in staurosporine range from 2.2h to 4.2h.

2. Figure 3e: Kap12 is not a 60S biogenesis protein. Dbp2 is a transcription factor, Rli1 is involved in 40S assembly, and Hep3 does not seem to be an actual gene in yeast.

We apologise for the reference to these genes and thank the reviewer for noticing this. We fixed the typos (KAP12 is KAP123 and HEP3 to HEF3).

3. The authors never probe translation, just polysome association. It is possible that polysome association increases because translation is slowed. They need to be careful in their wording. This also extends to the name "translation index". It should be called polysome binding index or something like that.

We agree with the reviewer, and we used "polyribosome association" instead of "translation" whenever possible. In addition, we clearly defined the term translation index in the figure legend as an indicator of polyribosome association.

4. I would suggest they rename the uL30B-5'A isoform to: uL30B-A-Nt (A-N-TERMINUS), as they are indicating it is the protein, not the mRNA that matters.

The changes are made as requested.

5. It is unclear how uL30B alike differs from uL30B (or uL30B-5'A).

The original names reflected the nature of the mutations produced. uL30BA-Like indicated the mutation of the 4 amino acids that differentiate the B from the A form to make the B sequence similar to that of A. In the case of uL30B-5'A, the sequence coding for the first 44 amino acids of uL30B was exchanged with that of A. Both constructs produced proteins with otherwise identical amino acids and displayed the same phenotype; they only differ in their primary nucleic acid sequence. For clarity, we removed the redundant mutations and simplified the naming of the remaining strains as requested in point 4.

6. There are typos throughout the manuscript:

a. Abstract. Programed should be programmed, b. P.8 ribosome-associated mRNAs were sequenced. Not was., c. Figure 7d axis title; protein is misspelled

Thanks, we have fixed these.

7. Abstract: Here we demonstrate that gene duplication may serve. That's awkward. Either you demonstrate or it may be the case. In addition, the authors really only investigate one case uL30, and no generalizations should be drawn.

We modified the statement and made it specific to uL30.

8. P3, bottom: Ferretti&Karbstein discuss RP concentration as a source of apparent functional specialization and provide evidence from a literature analysis that changes in their expression levels can account for apparent "resistance" effects.

We are aware of this publication and made sure from the start of our study to consider only strains that do not have decreased uL30 mRNA or protein levels. We also believe that the comparison of the two paralogs when expressed from identical promoters and the addition of the cross strains comparison in Supplementary Fig. 17 raises the quality of this work far ahead of the previous literature discussed in that publication.

9. P6. Secondary structure differences are not obvious from Figure S1.

We apologize for the missed reference. The phrase was corrected to refer to the tertiary structure showing Supplemental Fig. 2b.

10. P7. Staurosporine also reduces the polysomes in cells expressing uL30A and B. also, the paralog choice does affect the levels of other RPs. From both subunits.

We clarified the text as requested.

11. Title and P.17: most N-acetylation is co-translational, not posttranslational. Indeed, it is the point of the manuscript that changes are co-translational

We corrected the text as requested.

12. Figure 4: 60S associated mRNAs are not a thing.

We changed the statement to mRNA that co-sediments with the 60S.

13. Figure 7 is extremely difficult to follow. The legend does not explain much.

We have expanded the legends to better explain the model shown in Figure 7.

Reviewers' Comments:

Reviewer #1:

Remarks to the Author:

The authors have addressed all of my concerns. I believe that this very interesting and groundbreaking manuscript should be ready to be published.

Reviewer #2:

Remarks to the Author:

This is a revised version of a previously submitted manuscript by Malik-Ghulam et al. The revised version addresses some of my previous concerns, but not all, and many critical questions are still open. I do think that the data is presented a lot more clearly. Moreover, some of the new data raise significant new questions and actually demonstrate that the authors previous analyses were misleading. This manuscript still requires substantial revision.

1. The authors have measured ribosome abundance in all strains, but not with and without staurosporine. In particular, for the very clean data in Figure 5, we still do not know what ribosome abundance looks like in the presence of staurosporine. This is critical. In fact, their data demonstrates that in the AA and BB strains the minor isoforms becomes the major isoform after growth in staurosporine. This should be clarified and specifically pointed out...It is a general point that is important for not just this story on paralogs but many others. Just bc one isoform is minor in rich media doesn't mean it is minor always, and abundance can easily be behind different phenotypes.
2. They should also present the mRNA data under those conditions. This is especially critical for the data in Figure 5. Non-synonymous codons can have effects on mRNA levels as recently shown, and presumably that could be differential under different conditions.
3. There seems to be a lot of variation in the growth rates, especially in the presence of staurosporine....the authors should comment on that.
4. The analysis of the N-terminal acetylation requires information about the number of peptides that are observed in the mass spectrometry as well as a statistical analysis. I should have requested this earlier, and apologize for bringing this up only now, but a recent mass spec analysis we have done made it clear just how critical this is. I am very concerned that this analysis is based on 2-3 peptides and therefore entirely unreliable. In our own hands it is difficult to get statistically significant data for the very smallest RPs, and in particular for small peptides. Looking at Rpl30, the peptide should be 4 aa, which is probably poorly recovered, which explains the acetylation % of 0 and 100%, which is unlikely when one has dozens of peptides. I strongly suspect that the acetylation difference is not supported by current data.
5. The authors argue that differential expression of certain proteins underlies the staurosporine resistance. They have mass-spec data. They should look for the proteins whose mRNAs are changed in translational index.
6. Some of the effects are literally "created" by the normalizations that the authors perform. These MUST go away, and the resulting text must be revised. I am wondering about the growth data for that (see above), but currently the most glaring is the entire part about translation of long ORF mRNAs. This relies on the ratio of a long/short mRNA pair. First, the short mRNA is so short that it probably actually cannot accommodate the 7+ mRNAs in the P7 fraction. But even besides that, looking at the raw data for each mRNA that the authors now present in Figure S12a, it is clear that ul30aa is pretty much the same as ul30bb for the long ORF. The differences are in mRNAs in the 40S and 60S fractions, and their relevance is at best questionable. For the short mRNAs again, there are essentially

no differences EXCEPT in the very high polysome fractions. This I would argue is questionable due to the short length, which cannot even accommodate 7+ ribosomes. We have also often observed (when measuring each polysome fraction individually), that the last fraction of the gradient is different (e.g. polysome occupancy is low for the preceding two fractions, but then high in that fraction). We therefore exclude it from analysis, because we do not think it actually represents a translated mRNA. I have recently heard a discussion that it might be stress granules. So, the raw data actually say that it is short mRNAs that are differentially affected, and I am questioning that, because it is restricted to the questionable gradient fractions.

7. Because of this analysis the data in Supplemental 16h also need to be shown as raw data, not this weird normalization. The authors should also indicate the length of the ORF and not make me look it up.

8. The authors argue that the ORF length does not correlate with average translation index. If this is true, then it leads me to question their translational index data, as this correlation is well documented (see Weinberg and Bartel, Cell Rep 2016). However, I think this arises from the narrow focus on a sliver of genes the authors present (extremely long and short ones). In fact, the long genes they are inspecting are far longer than the average size of differentially affected mRNA according to their translational index. This analysis must be redone.

9. The authors should show the distribution of TE values in wt (from published data like the Weinberg paper or, less ideally, their translational index data) for all mRNAs and the ones up and downregulated in the AA, BB and clean S28-A/B strains. And see if the histograms are changed or not. This would show (or rule out) whether the mRNAs upregulated are simply the ones otherwise poorly translated. I am concerned that this is behind this story.

10. I had asked for an experiment that tests if other means of translational inhibition also produced staurosporine resistance. That experiment is still missing.

11. Similarly, I had suggested a NAT k/o. the authors have not done this experiment and argue it might not work. But it is not hard. And the converse experiment, NAT overexpression should rescue the effects from the A isoform...

12. The authors have taken out critical data that show the broader effects of these genetic manipulations. These are really important, not just for this study, but other similar ones, and they should be in the manuscript, together with a critical acknowledgement of these effects. I am thinking about what used to be Figure S2g.

13. The authors should provide a Table that shows the mRNAs opposite regulated by the AA and BB strains. This is impossible to figure out from table S3.

14. L. 236-238: This statement may be incorrect as data are only presented without staurosporine.

Reviewer #2 (Remarks to the Author):

This is a revised version of a previously submitted manuscript by Malik-Ghulam et al. The revised version addresses some of my previous concerns, but not all, and many critical questions are still open. I do think that the data is presented a lot more clearly. Moreover, some of the new data raise significant new questions and demonstrate that the authors previous analyses were misleading. This manuscript still requires substantial revision.

We thank the reviewer for their time and are glad they find the data clearer. We also appreciate the reviewer's interest in the subject, but we obviously take issue with labelling our earlier analysis to be misleading. We made every effort to explain how the analyses were made and the limitation of the experiments were discussed to give the reader every opportunity to evaluate and interpret the data. Regardless, of this difference in opinion, we are grateful for the time the reviewer put into reviewing this work and made every effort to accommodate their concerns.

1. The authors have measured ribosome abundance in all strains, but not with and without staurosporine. In particular, for the very clean data in Figure 5, we still do not know what ribosome abundance looks like in the presence of staurosporine. This is critical. In fact, their data demonstrates that in the AA and BB strains the minor isoforms becomes the major isoform after growth in staurosporine. This should be clarified and specifically pointed out...It is a general point that is important for not just this story on paralogs but many others. Just bc one isoform is minor in rich media doesn't mean it is minor always, and abundance can easily be behind different phenotypes.

As requested, we have now included the measurement of rRNA abundance after exposure to staurosporine in new supplementary Figure 19b. As indicated, the mutations of uL30 paralog do not change the abundance of rRNA as compared to wild type cells in staurosporine. Once again, these results show that difference in the ribosome number is not the basis of uL30 paralog effect.

2. They should also present the mRNA data under those conditions. This is especially critical for the data in Figure 5. Non-synonymous codons can have effects on mRNA levels as recently shown, and presumably that could be differential under different conditions.

As requested, we included new data in Supplementary Figure 19c showing the effect of staurosporine on the abundance of RPL7 mRNA. The data show that exposure to staurosporine do not change mRNA abundance in response to staurosporine when the genes are expressed from plasmid supporting the data in Figure 5. This clearly indicates that differences in the effects of the plasmid borne copies of uL30 paralog is not due to changes in the abundance of uL30 mRNA. We only observed staurosporine dependent increase in mRNA abundance when the chromosomal gene of uL30A is the sole source of uL30 in the cells (uL30a Δ and uL30bb) consistent with the cell requiring this protein

version when treated with staurosporine. This response seems to be specific to uL30B since the opposite is observed when a single uL30A (uL30b Δ) is the sole chromosomal source of uL30.

3. There seems to be a lot of variation in the growth rates, especially in the presence of staurosporine....the authors should comment on that.

The variation in growth rate in the presence of staurosporine is expected given the inherent difficulty in controlling small variation in drug uptake. These fluctuations are normally generated by differences in the age of the cells, small differences in the saturation of the overnight culture, and slight changes in growth temperature. In almost all cases, growth after exposure to any drugs is much more variable than in untreated cells. However, despite these difficulties our reported growth is well within the accepted norm and does not change the experiment's conclusion. Indeed, the average relative variation for the doubling times reported in Supplementary Figure 18 is 4,82% for untreated cells and 10.94% for cells treated with staurosporine. The most important point here is that these small changes do not in any way affect or change the overall conclusion or the obvious difference between the paralog effects, which is the main take home message here.

4. The analysis of the N-terminal acetylation requires information about the number of peptides that are observed in the mass spectrometry as well as a statistical analysis. I should have requested this earlier, and apologize for bringing this up only now, but a recent mass spec analysis we have done made it clear just how critical this is. I am very concerned that this analysis is based on 2-3 peptides and therefore entirely unreliable. In our own hands it is difficult to get statistically significant data for the very smallest RPs, and in particular for small peptides. Looking at Rpl30, the peptide should be 4 aa, which is probably poorly recovered, which explains the acetylation % of 0 and 100%, which is unlikely when one has dozens of peptides. I strongly suspect that the acetylation difference is not supported by current data.

In the case of paralog specific N-terminal acetylation, only one peptide could be considered for paralog specific analysis, given the position of the modification. Since the paralogs differ in their 2nd and 3rd amino acids, we can distinguish between them using the N-terminal peptide. As well noted, the expected N-terminal peptide for uL30 would be 4 amino-acids long and would be below the detection range of the analyzer used for this study that can detect peptides with m/z ranging from 350-1600. For this reason, we have considered the 13 amino acid long peptide that results from the cleavage at the second lysine of uL30. The analysis took into consideration partial cleavage, and we verified that the efficiency of LysC digestion was similar across the samples at the proteome level. In addition, we also compared the average peptide intensity detected across the proteome to that of all uL30 peptides and the uL30 N-terminal peptides to ensure that the N-terminal 13 amino acid long peptide was not poorly detected compared to the rest of the proteome. Most importantly, we found very little variation between experiments. In all cases the N-

terminal acetylation of uL30A in the 4 wt samples varied from 9,1% to 12% between experiments whereas no unacetylated uL30B was detected. Therefore, it is clear that the changes we see in acetylation is stable and do not vary between experiments and that the differences between wild type cells, mutated cells, treated and untreated cells are all statistically significant. For clarity, we included more details about the method used for the detection of uL30 N-terminus in the methods section.

5. The authors argue that differential expression of certain proteins underlies the staurosporine resistance. They have mass-spec data. They should look for the proteins whose mRNAs are changed in translational index.

To address the reviewer's comment, we included new data in Figure 4 c showing the protein abundance of the 50 genes exhibiting the biggest paralog dependent difference in translation.

6. Some of the effects are literally "created" by the normalizations that the authors perform. These MUST go away, and the resulting text must be revised.

Internal normalization is essential to avoid inter-experiment variation and misleading the reader because of innate changes in protein and mRNA amount, variation in extract and day to day variation. We strongly believe that the normalization increases rather than decreases accuracy. However, to accommodate the reviewer's request and be sure that the reader has all the information we also included all the raw data and furthermore included in the text references to the raw data to be absolutely sure that the reader has all the facts in hand.

I am wondering about the growth data for that (see above).

The growth data of all strains are normalized to wild type cells to control for small variations in media and culture condition. However, we do provide the non-normalized doubling time for all strains in Supplementary Fig. 18.

Currently the most glaring is the entire part about translation of long ORF mRNAs This relies on the ratio of a long/short mRNA pair. First, the short mRNA is so short that it probably actually cannot accommodate the 7+ mRNAs in the P7 fraction.

To address this reviewer comment we have now modified the analysis in Fig 4e to compare the association of the long and short mRNA in the different fractions relative to control mid-size mRNA Osh6, which could be detected in all fractions. In this way, we could compare the relative translation of long and short ORF while controlling for overall change in ribosome profile.

But even besides that, looking at the raw data for each mRNA that the authors now present in Figure S12a, it is clear that ul30aa is pretty much the same as uL30bb for the

long ORF. The differences are in mRNAs in the 40S and 60S fractions, and their relevance is at best questionable. For the short mRNAs again, there are essentially no differences EXCEPT in the very high polysome fractions. This I would argue is questionable due to the short length, which cannot even accommodate 7+ ribosomes. We have also often observed (when measuring each polysome fraction individually), that the last fraction of the gradient is different (e.g., polysome occupancy is low for the preceding two fractions, but then high in that fraction). We therefore exclude it from analysis, because we do not think it represents a translated mRNA. I have recently heard a discussion that it might be stress granules. So, the raw data actually say that it is short mRNAs that are differentially affected, and I am questioning that, because it is restricted to the questionable gradient fractions.

The reviewer's analysis did not take in account the fact that the overall translation is inhibited in ul30bb when compared to ul30aa strains. That is why the difference in the translation of long and short open reading frame is not detectable without normalization to control mRNA. The same level of mRNA association in ul30aa and ul30bb means that this mRNA is more translated than the rest of the mRNA in bb. To address this reviewer comment we have recalculated the data of Fig 4e using internal mid-size mRNA as described above. In addition, we had excluded the last fraction of the gradient, where the profile becomes blurry, from the analysis to address this reviewer concerns regarding possible co-sedimentation with other complexes. Furthermore, we made it clear that changing the expressed version of the uL30 paralog shifts the relative translation of long and short mRNA but does not result in a net increase in the overall translation of these mRNA.

7. Because of this analysis the data in Supplemental 16h also need to be shown as raw data, not this weird normalization. The authors should also indicate the length of the ORF and not make me look it up.

As explained for point 6, relative comparison is essential to remove the influence of overall variation in translation level and clarify that only the ratio of translated short and long ORF that changes. Accordingly, we improved the comparison by including an internal mid-size mRNA to control for the difference in the long and short ORF ribosome occupation potential. Furthermore, we also included the raw data in Supplementary Figure 17g, stating the length of the analyzed ORFs in the legend and made it clear in the text that it is the relative translation and not absolute translation that changes in these strains.

8. The authors argue that the ORF length does not correlate with average translation index. If this is true, then it leads me to question their translational index data, as this correlation is well documented (see Weinberg and Bartel, Cell Rep 2016). However, I think this arises from the narrow focus on a sliver of genes the authors present (extremely long and short ones). In fact, the long genes they are inspecting are far longer than the

average size of differentially affected mRNA according to their translational index. This analysis must be redone.

There is a clear misunderstanding here. The average translation index as stated in the text cannot be directly compared to translation efficiency (TE) generated by Weingerg and Bartel is based on ribosome density (size adjusted number of ribosome per-nucleotide) while the translation index referred to in our study simply indicates the proportion of each mRNA population associated with polyribosomes relative to that associated with monosomes and subunits. As such our translation index gives an indication of the proportion of each mRNA being engaged with translation and not the number of ribosomes per mRNA as in Bartel work. Indeed, we recalculated our data accounting for ORF lengths, and we now see an anti-correlation between the translation (as function of ribosome density) of long and short ORF as previously reported (See supplementary Fig 11). This is clearly, consistent with our model and hypothesis suggesting that uL30 paralog effect is related to translation elongation and the stability of mRNA association with ribosome. In order to address the reviewer's comment, we have now added a detailed comparison of the length effect on translation index and length adjusted translation index in Supplementary Figure 11. This figure includes analysis of all ORFs and not a sliver of genes as requested. The data clearly shows that long ORFs could be considered under translated only when ribosome density is factored in and not in terms of the proportion of mRNAs engaged with polyribosomes. Most importantly, the data indicate that long ORFs are more translated than short ORFs in the BB strains regardless of the methods of calculation. We mentioned the difference between the methods of calculation and the reference to Bartel's paper in page 14.

9. The authors should show the distribution of TE values in wt (from published data like the Weinberg paper or, less ideally, their translational index data) for all mRNAs and the ones up and downregulated in the AA, BB and clean S28-A/B strains. And see if the histograms are changed or not. This would show (or rule out) whether the mRNAs upregulated are simply the ones otherwise poorly translated. I am concerned that this is behind this story.

As explained in point 8, we added new supplementary Figure 11 that clearly indicates that differences in estimating the translation efficiency of long ORFs length depends on whether we factor in density or simply the percent RNA engaged with polyribosome. As to whether the translation efficiency is behind the translation upregulation in BB, the answer is partially yes if translation efficiency is calculated as factor of ribosome density or ribosome number per mRNA molecule. On the other hand, the answer is clear no if we only considered the proportion of mRNA associated

with polyribosome. We made this clear in the new comparison we included in Supplementary Figure 11. The link between the long ORFs and ribosome density directly supports our hypothesis that uL30/RPL7 paralogs affect translation elongation, and the time ribosomes remain associated with the mRNA. That is said it is clear that ribosome density is not the only factor since the effect of uL30 paralog do not increase as the translation index decrease (Supplementary Figure 11b). Indeed, comparing short and long mRNA, with the same length adjusted translation index, continues to show that long ORFs with average ribosome density could be more translated in BB. We explained this nuance in the text and explained that while genes with normally low ribosome density are more likely to be over translated in BB, which according to our model consistent with uL30B paralog propensity to promote ribosome association with mRNA, it is not the only factor and long genes could be more translated even when having the ribosome density as shorter genes. Moreover, we reproduced the same analysis with strains expressing the paralogs from plasmids in Supplementary Figure 11 and show that the same trend is maintained. We also added Supplementary Figure 12 to clearly demonstrate that over and under translation of genes in ul30bb is not strictly linked to length. We always detect increased translation of long genes even in genes with above average translation index in wild type cells. Notably, the percentage of mRNAs that are over translated in ul30bb is almost constant for the 60% of the mRNAs having “middle” WT translation index. The reader now has all the facts in hand and the conclusions are balanced to accommodate for the possible difference in the interpretation of the direct experimental evidence.

10. I had asked for an experiment that tests if other means of translational inhibition also produced staurosporine resistance. That experiment is still missing.

We have added new data in Supplementary Figure 4 c and d showing that the deletion of translation elongation factors that slow elongation (Tef4) and not those affecting initiation (eIF2A or Tif4632) results in staurosporine resistance. This data fully supports our model suggesting that resistance of staurosporine is produced by paralog dependent attenuation of translation elongation.

11. Similarly, I had suggested a NAT k/o. the authors have not done this experiment and argue it might not work. But is is not hard. And the converse experiment, NAT overexpression should rescue the effects from the A isoform...

The problem we have with this suggestion is not in the experimental complexity but rather its pertinence and rationale. This is an experiment that will lead to potentially misleading interpretation regardless of what results we obtain. Changed resistance to staurosporine after NAT knockout does not necessarily prove that the acetylation of uL30B is necessary since the lack of acetylation of other proteins might be the cause of the phenotype. On the other hand, the absence of effects or increased resistance to staurosporine would not prove that the acetylation of uL30 is not required since changes in the modification of

other protein may annul the effect of uL30 acetylation. The over expression also does not prove or disprove the role of uL30 acetylation to staurosporine tolerance since it will alter the overall acetylation of the entire proteome. Indeed, as part of another study we have data showing that many ribosomal proteins with different and opposing effects on translation are acetylated. In addition, the yeast NAT complex includes 5 subunits and the deletion of each of these genes has different phenotype. Deletion of Nat1, which is one of the most studied, results in slow growth, failure to enter stationary phase, and defects in sporulation, modifies position effect at telomere (PMID: 2551674, PMID: 1600941 PMID: 14517307, PMID: 2681143). Overexpression of NAT 1 can lead to chromosomal instability and modify chromatin structure and function. How could one interpret the effect of this deletion and relate it to a single amino acid acetylation in one protein? This experiment can only bring confusion to the issue at hand. Indeed, the contribution of acetylation to ribosome biology is the subject of a full separate manuscript and it could not be adequately and correctly addressed with all the needed controls within the scope of this study.

12. The authors have taken out critical data that show the broader effects of these genetic manipulations. These are really important, not just for this study, but other similar ones, and they should be in the manuscript, together with a critical acknowledgement of these effects. I am thinking about what used to be Figure S2g.

Figure S2g was removed because it was confusing to the readers and the reviewers. It was addressing the effect of uL30 on paralog pairs ratios and not total protein amount. As such it did not add to the main message or identify any significant differences between paralogs. To facilitate the understanding of the figure and its conclusion replaced S2g with Supplementary Figure 2f that compares the changes in total protein and mRNA amount.

13. The authors should provide a Table that shows the mRNAs opposite regulated by the AA and BB strains. This is impossible to figure out from table S3.

We included the new table S5 as requested.

14. L. 236-238: This statement may be incorrect as data are only presented without staurosporine.

The statement is correct. It simply states that cells expressing the same number of ribosomes under normal condition may still show difference in staurosporine resistance. In any case, we now also added the data showing the ribosome numbers of cells growing in the presence of staurosporine which further strengthen the accuracy of the sentence (see Supplementary Figure 19).